# Netrin-1 and its receptor DCC modulate survival and death of dopamine neurons and Parkinson's disease features

Mélissa Jasmin[1],[†], Eun Hee Ahn[2],[†] (ID), Merja H Voutilainen[3],[4],[†], Joanna Fombonne[1], Catherine Guix[1], Tuulikki Viljakainen[3],[4], Seong Su Kang[2], Li-ying Yu[3], Mart Saarma[3],[‡], Patrick Mehlen[1],[*],[‡] (ID) & Keqiang Ye[2],[*],[‡] (ID)

## Abstract

The netrin-1/DCC ligand/receptor pair has key roles in central nervous system (CNS) development, mediating axonal, and neuronal navigation. Although expression of netrin-1 and DCC is maintained in the adult brain, little is known about their role in mature neurons. Notably, netrin-1 is highly expressed in the adult substantia nigra, leading us to investigate a role of the netrin-1/DCC pair in adult nigral neuron fate. Here, we show that silencing netrin-1 in the adult substantia nigra of mice induces DCC cleavage and a significant loss of dopamine neurons, resulting in motor deficits. Because loss of adult dopamine neurons and motor impairments are features of Parkinson's disease (PD), we studied the potential impact of netrin-1 in different animal models of PD. We demonstrate that both overexpression of netrin-1 and brain administration of recombinant netrin-1 are neuroprotective and neurorestorative in mouse and rat models of PD. Of interest, we observed that netrin-1 levels are significantly reduced in PD patient brain samples. These results highlight the key role of netrin-1 in adult dopamine neuron fate, and the therapeutic potential of targeting netrin-1 signaling in PD.

**Keywords** netrin-1/ DCC; neurorestoration; Parkinson's disease
**Subject Categories** Molecular Biology of Disease; Neuroscience
**The EMBO Journal (2021) 40: e105537**

## Introduction

The canonical guidance cue netrin-1 is a laminin-related secreted protein now emerging as a multifunctional plasticity cue (Kennedy *et al*, 1994; Serafini *et al*, 1994; Sun *et al*, 2011). Its main receptors,

Deleted in Colorectal Cancer (DCC) and UNC5B, belong to the class of dependence receptors, a functional family of receptors shown to be cleaved and to trigger cell death in settings of poor ligand availability (Mehlen *et al*, 1998; Llambi, 2001). Hence, these receptors exhibit a dual signaling depending on the cellular context. Typically, when unbound to their ligands, dependence receptors undergo proteolysis through caspase cleavage, which in turn activates cell death, whereas binding to their ligands induces a "positive" signaling ensuring cell survival but also the activation of the ligand's canonical pathway (proliferation, migration, differentiation for instance; Negulescu & Mehlen, 2018). Along this line, netrin-1 was shown to act as a neuronal survival cue in various contexts (Furne *et al*, 2008; Tang *et al*, 2008).

Despite extensive studies on the role of netrin-1 in the development of the mammalian central nervous system (CNS), less is known about its role in adulthood. Indeed, both netrin-1 and DCC continue to be expressed in the adult CNS, particularly in the substantia nigra (Livesey & Hunt, 1997). This midbrain structure, involved in motor control, consists of two parts, the *pars compacta* (SNpc) and the *pars reticulata* (SNpr). The former is mainly composed of dopamine (DA) neurons while the latter of GABAergic neurons. Netrin-1 receptor, DCC, is highly expressed in DA neurons of the SNpc (Volenec *et al*, 1998; Osborne *et al*, 2005; Reyes *et al*, 2013). These neurons supply the dorsal striatum with dopamine and are known to be particularly vulnerable to degeneration in Parkinson's disease (PD), the second most common neurodegenerative disease. The preferential and progressive degeneration of DA neurons of the SNpc in PD leads to debilitating motor symptoms heavily impacting the quality of life of patients with PD. To date, the etiology of the disease is still obscure and current therapies are symptomatic and do not prevent the progression of the disease. Thus, understanding what makes these neurons particularly vulnerable to degeneration and what could promote their resistance may open ways to identifying new targets for the treatment of the disease.

1  Apoptosis, Cancer and Development Laboratory – Equipe labellisée 'La Ligue', LabEx DEVweCAN, Centre de Recherche en Cancérologie de Lyon, INSERM U1052-CNRS UMR5286, Centre Léon Bérard, Université de Lyon, Université de Lyon1, Lyon, France
2  Department of Pathology and Laboratory Medicine, Emory University School of Medicine, Atlanta, GA, USA
3  Institute of Biotechnology, HiLIFE, University of Helsinki, Helsinki, Finland
4  Division of Pharmacology and Pharmacotherapy, Faculty of Pharmacy, University of Helsinki, Helsinki, Finland
   *Corresponding author. Tel: +33 4787828; E-mail: patrick.mehlen@lyon.unicancer.fr
   **Corresponding author. Tel: +1 404 712 2806; E-mail: kye@emory.edu
   †These authors contributed equally to this work
   ‡These authors contributed equally to this work as senior authors

Given DCC expression in adult nigral dopamine neurons and its reported role in cell death as a dependence receptor, we interrogated the contribution of netrin-1 in the fate of these neurons.

## Results

### Netrin-1 is expressed in the substantia nigra and localizes with SNpc DA neurons

To begin to address the contribution of netrin-1 in SN DA neurons, we first confirmed its expression in the adult mammalian substantia nigra. According to the Allen Human Brain Atlas data bank, *netrin-1* expression in the adult brain is globally low except in some structures of the brainstem. Especially, netrin-1 expression levels are among the highest in the substantia nigra (SN; Fig 1A). Using an S-gal gene reporter method, we found that *netrin-1* expression in the mouse brain was, as in the human brain, particularly strong in the SN (Fig 1B) as previously reported (Livesey & Hunt, 1997). We then sought to look at netrin-1 protein localization in the nigrostriatal pathway. Tyrosine hydroxylase (TH) was used as a marker for DA neurons. We found that, like its receptors (Appendix Fig S1A), netrin-1 was present in the SNpc and largely colocalized with ventral tier DA cell bodies (Fig 1C). Of interest, ventral tier SNpc DA neurons are reportedly the most vulnerable to degeneration in PD (Fearnley & Lees, 1991). Since these neurons send axons to the striatum and since DCC is also expressed at dopamine axon terminals (Osborne *et al*, 2005), we checked whether netrin-1 could be secreted along dopamine axons terminals (Dopamine transporter (DAT) staining). Netrin-1 immunoreactivity was diffuse and barely noticeable throughout the striatum (Fig 1D), except for a small population of netrin-1-expressing cells, presumably interneurons (Shatzmiller *et al*, 2008). Netrin-1 from striatal interneurons may have a paracrine action on DCC expressing axon terminals.

### Netrin-1 loss induces nigral dopamine cell death

In the SNpc, the high expression of netrin-1 suggests that it may contribute to the function or maintenance of SNpc DA neurons in adult, possibly by an autocrine loop. To test this, we used a loss-of-function approach conditionally knocking-out *netrin-1* specifically in the adult SN. Netrin-1 depletion was achieved injecting stereotaxically and unilaterally AAV6-Cre-eGFP virus vector (Cre) into the SN of 3-month-old netrin-1$^{fl/fl}$ mice (Fig 2A and B and Appendix Fig S1B and C, Cre panel). The levels of netrin-1 and TH in the SN and mice motor behavior were monitored six weeks after *netrin-1* conditional knockout (KO) in the SN. The loss of TH and VMAT2-positive phenotype in nigral cell bodies and in striatal fibers (Fig 2C and Appendix Fig S1D and E) coupled with a massive increase of TUNEL reactivity in the SNpc (Fig 2D) demonstrate that the conditional deletion of *netrin-1* elicited the death of nigral DA neurons. Consistently, *netrin-1* KO eventually led to significant motor impairments as evaluated by rotarod, cylinder, and grid motor behavior tests (Fig 2E). Interestingly, netrin-1 deletion in the VTA did not result in significant loss of dopamine neurons (TH-positive; Fig 2B) as compared to dopamine loss in the SN. Thus, netrin-1 appears to be required for nigral DA neuron maintenance in adult animals. Moreover, the loss of netrin-1 was associated with increased levels of alpha-synuclein, a protein which accumulates and forms toxic aggregates in PD (Fig 2F) and increased DCC and UNC5B receptors levels and cleavage (Fig 2F). Similarly, the titration of netrin-1 using biologics (Paradisi *et al*, 2009; Broutier *et al*, 2016) in primary neurons induced an increase of DCC and UNC5B protein levels and cleavage (Appendix Fig S2A and B) that was associated with increased neuronal death (Appendix Fig S2C). Blocking caspases' activation by using caspases inhibitors prevented netrin-1 deprivation-induced DCC or UNC5B proteolytic cleavage and loss of tyrosine hydroxylase (TH) levels (Appendix Fig S2D).

### Unbound DCC triggers dopamine cell death

Receptor cleavage by caspases and activation of cell death in the absence of ligand is a feature of dependence receptors. To investigate whether netrin-1 receptors could actively mediate dopamine cell loss in the absence of netrin-1, we co-injected, into the ventral midbrain of netrin$^{fl/fl}$ mice, AAV6-Cre-eGFP virus vector with lentiviral ShRNA constructs against *Dcc* (ShDCC) or *Unc5b* (ShUNC5B) which have no effect on neuron survival *in vitro* (Appendix Fig S2E and F). Immunofluorescence (Fig 3A and B) and TUNEL (Fig 3C) analyses demonstrated that knocking down *Dcc* in the SN of netrin$^{fl/fl}$ mice partially protected dopamine neurons in the absence of netrin-1 whereas knocking down *Unc5b* did not prevent dopamine cell death as strongly as *Dcc*. Similar observations were made from SN lysates: DCC depletion, and at a much lesser extent UNC5B, partially prevented the loss of TH and active caspase-3 protein levels (Fig 3D, left panels). Consistently, motor behavior tests demonstrated that knocking down either *Dcc* or *Unc5b* attenuated motor deficits induced by netrin-1 conditional depletion in the SN (Fig 3E). Of note, DCC depletion exhibited more prominent protective effect compared to UNC5B depletion. An explanation might be that, unlike DCC, UNC5B seems to be poorly expressed in the substantia nigra. However, we cannot exclude that UNC5B alone may be less efficient in inducing cell death than DCC or that both receptors may be required. Taken together, our results indicate that the loss of netrin-1 in the SN induces dopamine neurons death, at least in part, via unbound DCC.

### Netrin-1 is neuroprotective and neurorestorative in models of PD

Next, we investigated whether, conversely, netrin-1 gain of function could protect or rescue SN dopamine neurons from degeneration. To this end, we tested the effect of netrin-1 overexpression on the degeneration of dopamine neurons in netrin-1 inducible transgenic mice lesioned intrastriatally and unilaterally with 6-hydroxydopamine (6-OHDA). 6-OHDA brain lesion has been widely used to model PD and to screen for neurorestorative molecules. When injected into the striatum, this toxin is captured by DAT and auto-oxidizes in the cell causing oxidative stress and mitochondrial disfunction leading to the sensitization of neurons and eventually neuronal death within 4 weeks. Injected unilaterally, 6-OHDA eventually causes an imbalance in DA activity between the two hemispheres due to the degeneration of neurons in only one hemisphere. Amphetamine administration exacerbates this imbalance and induces so-called "ipsilateral rotations" which intensity correlates with the extent of degeneration. Amphetamine-induced ipsilateral rotations are thus used as a read-out to follow the effect of molecules on dopamine neuron degeneration (lesion). Netrin-1 inducible

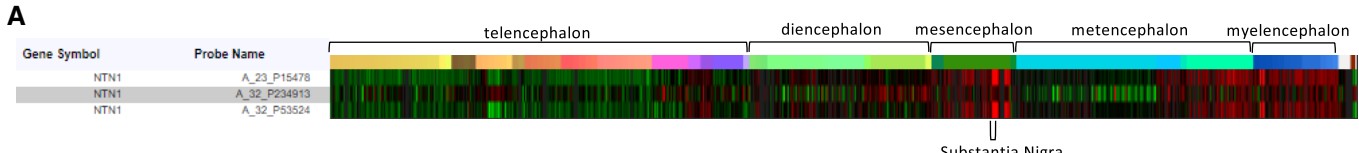

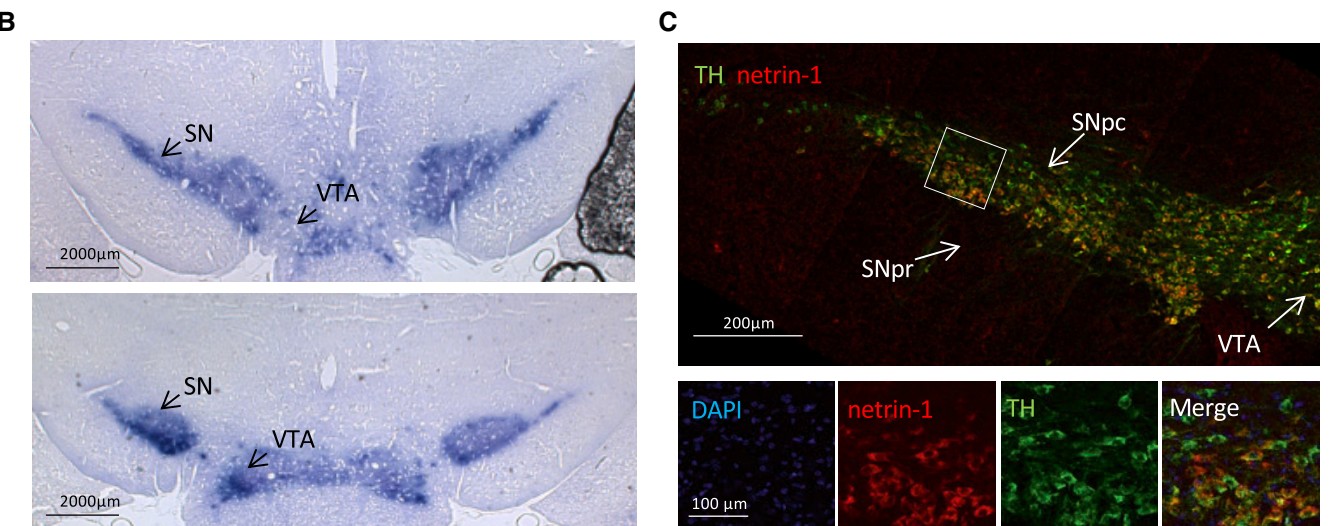

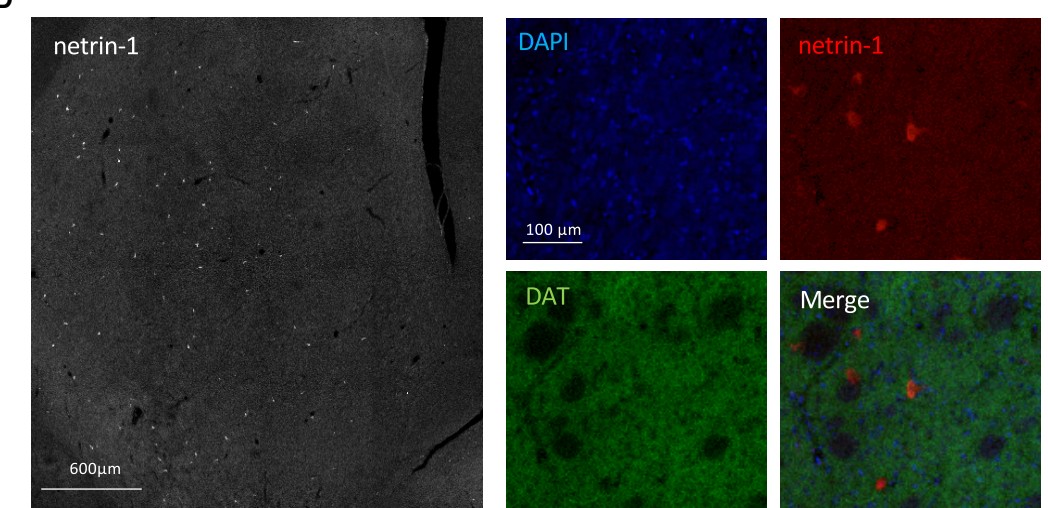

**Figure 1. Netrin-1 is highly expressed in the substantia nigra and colocalizes with SNpc DA neurons.**

A  Netrin-1 (NTN1) gene expression profiling from six adults control brains using microarray data from the Allen Human Brain Atlas.

B  Reporter gene expression in the midbrain (substantia nigra (SN) and ventral tegmental area (VTA)) of netrin-1[+/lacZ+] adult (P60) mice incubated with S-Gal. (Scale bar, 2,000 μm). Two coronal sections of the mice midbrain are shown from the more anterior is shown in the upper panel.

C  Netrin-1 protein localization in the SN of adult rat. (Scale bar, 200 μm). Upper panel, confocal image of the SN. Higher magnification of the SNpc (bottoms panels). Netrin-1 (red), tyrosine hydroxylase (TH) (green).

D  Localization of netrin-1 protein in the adult rat striatum. (Scale bar, 600 μm). Left panel, confocal image of the rat striatum. Netrin-1 (red), dopamine transporter (DAT) (green). Right panels, higher magnification.

Data information: All images are made from coronal sections, and images are oriented with the dorsal part of the tissue at the top, and ventral part at the bottom.

mice received intraperitoneal (i.p.) injection of tamoxifen (see Methods) two weeks after 6-OHDA lesion to induce Netrin-1 overexpression. Amphetamine-induced ipsilateral rotation assays were performed right before tamoxifen injections (at week 2) and a month later (at week 6) (Fig 4A and Appendix Fig S3A). We found that ipsilateral rotational behavior was markedly reduced in netrin-1

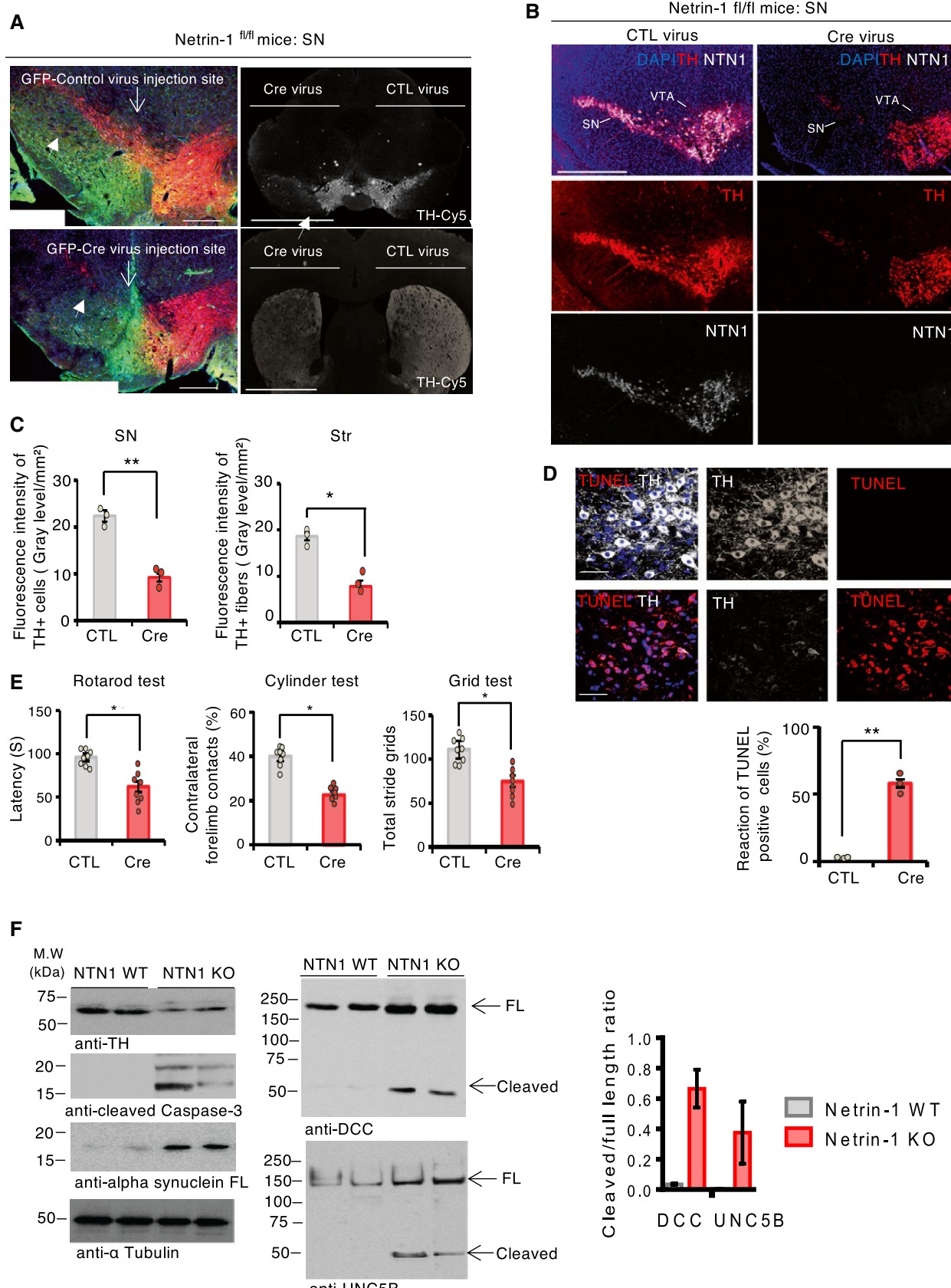

**Figure 2.**

**Figure 2. Silencing netrin-1 in the SN elicits TH loss and motor impairment.**

A   Representative images of dopamine neurons (TH-positive) in the substantia nigra (SN) (left panel) and in the SN and striatum (right panel top, and bottom, respectively) by immunofluorescence, six weeks after the intranigral injection of control (CTL) (AAV6-GFP) or Cre (AAV6-GFP Cre) virus vectors in Netrin-1$^{fl/fl}$ mice (left panel: green-GFP, red-TH; right panel: Cy-5-TH) (Scale bar, 1,000 μm). Filled arrows indicate SN TH dopamine neurons.

B   Representative images of TH and netrin-1 stainings by immunofluorescence (scale bar, 2,000 μm) six weeks after the intranigral injection of CTL adenovirus vector (left panel) or Cre adenovirus vector (right panel), in the midbrain (substantia nigra (SN) and ventral tegmental area (VTA) of Netrin-1$^{fl/fl}$ mice.

C   Quantification of fluorescence intensity of TH-positive neurons, in the substantia nigra (SN) (left), and fibers, in the striatum (str) (right), six weeks after adenovirus vector injection, in Netrin-1$^{fl/fl}$ mice. Bars and error bars represent the mean ± SEM. Statistical significance was determined by an unpaired *t*-test. *$P < 0.05$; **$P < 0.01$. $N = 3$ each group.

D   Terminal deoxynucleotidyl transferase (TdT) dUTP Nick-End Labelling (TUNEL) reactivity in the SN after Cre and control virus vector injection, in Netrin-1$^{fl/fl}$ mice (Scale bar, 50μm). The apoptotic index is expressed as a percentage of TUNEL-positive neurons out of the total number of TH-positive neurons. Bars and error bars represent the mean ± SEM. Statistical significance was determined by an unpaired *t*-test. **$P < 0.01$. $N = 3$ each group.

E   Motor behavior assays at 6 weeks after treatment. $N = 8$ animals/group. Bars and error bars represent the mean ± SEM. Statistical significance was determined by an unpaired *t*-test. *$P < 0.05$

F   Immunoblot of SN lysates from netrin-1 wild type (WT) and netrin-1 conditional KO mice (left upper panel). Densitometry band quantification of cleaved receptor over full-length receptor ratio (lower right panel). We randomly selected two mice from each group for immunoblot analysis. $N = 3$ independent experiments. Bars and error bars represent the mean ± SEM. Statistical significance was determined by an unpaired *t*-test. *$P < 0.05$; P**$< 0.01$.

Source data are available online for this figure.

overexpressing (O/E) mice compared to control littermates (Fig 4B) six weeks after 6-OHDA lesion. Immunohistochemical analyses showed that TH-positive striatal fiber density at 6 weeks was significantly higher in the netrin-1 O/E group than in netrin-1 WT group indicating that netrin-1 induction restored dopamine axonal projections in the striatum (Fig 4C). The number of TH-positive cells in the SNpc at 6 weeks was also higher in netrin-1 O/E group compared with netrin-1 WT group, although this was not significant (Fig 4D). These results suggest that netrin-1 is functionally neurorestorative in this 6-OHDA unilateral mouse model of PD.

Despite its extensive use to model PD, 6-OHDA lesion fails to recapitulate Lewy body inclusions and alpha-synuclein (SNCA) toxic aggregation, which are a hallmark of PD. However, most of "alpha-synuclein" based PD animal models have been difficult to reproduce and fail to show proper loss of SN dopamine neurons and associated motor deficits. Therefore, to test the effect of netrin-1 on alpha-synuclein-induced toxicity, we used 3-month-old human SNCA transgenic mice injected intraperitoneally with 1-methyl-4-phenyl-1,2,3,6-tetrahydropyridine (MPTP), a neurotoxin inducing an acute (within 4–6 days) degeneration of dopamine neurons, to potentiate alpha-synuclein toxicity. Since MPTP induces an acute degeneration of dopamine neurons, we chose to inject recombinant human netrin-1 shortly before MPTP i.p. injection (Fig 4E). As shown in Fig 4F and G, the MPTP injection in WT mice elicited an acute loss of SN DA neurons (TH-positive cells) that was not

rescued by netrin-1 treatment suggesting that netrin-1 is not a general inhibitor for inflammation-associated neuronal cell death. However, the decrease of TH-positive cells and fibers in response to MPTP injection is significantly less pronounced in netrin-1-treated SNCA Tg mice than in non-treated SNCA Tg mice, demonstrating that one single injection of recombinant netrin-1 significantly inhibits the additional loss of DA neurons induced by SNCA expression. Interestingly, netrin-1 treatment in the SN was associated with decreased serine 129 (S129) phosphorylated alpha-synuclein, an alleged pathological form of alpha-synuclein which promotes the formation of alpha-synuclein toxic aggregates (Anderson *et al*, 2006; Appendix Fig S3B). The rescue of TH-positive cells in netrin-1-treated mice is associated with behavioral benefits as measured by rotarod test (Fig 4H), cylinder test (Fig 4I), or Grid test (Fig 4J).

Taken together, these mice models indicate that recombinant human netrin-1 may be a potent therapeutic biologic for PD. Indeed, local brain injection of recombinant trophic factors appears as a credible therapeutic option for PD as exemplified with the current clinical trials assessing GDNF and CDNF (NCT03652363 and NCT03775538). We thus explored further the therapeutic potential of exogenous netrin-1 administration using the unilateral 6-OHDA rat model, one of the most extensively used preclinical PD models.

Rats received stereotaxic injections of 6-OHDA into three sites of the right striatum as described in Penttinen *et al* (2016) to induce a progressive and non-reversible degeneration of SN DA neurons.

**Figure 3. Unbound DCC participates to netrin-1 depletion-induced dopamine cell loss.**

A   Representative images of TH + dopamine neurons and fibers staining by immunofluorescence, in the SN and the striatum (Str) 6 weeks after CTL (AAV6-GFP); Cre (AAV6-GFP Cre); sh DCC (AAV6-GFP Cre + sh DCC); and shUNC5B (AAV6 GFP Cre + Sh-UNC5B) injection into the SN of Netrin-1$^{fl/fl}$ mice. (Scale bar, 2,000 μm).

B   Quantification of TH fluorescent intensity in the SN (upper bar graph) and striatum (Str) of Netrin-1$^{fl/fl}$ mice (lower bar graph). Data are shown as mean + SEM. Statistical significance was determined by an unpaired *t*-test. $N = 3$ each group. Tukey *post hoc* analysis after one-way ANOVA *$P < 0.05$, **$P < 0.01$, N.S., not significant.

C   DCC depletion mitigates netrin-1 depletion-induced apoptosis in Netrin-1$^{fl/fl}$ mice. TH-positive cell loss was assessed by TUNEL assay in the SN. Upper panel, TUNEL (red) and TH (Cy5-white) (Scale bar, 50 μm). Apoptotic index (bar graph, bottom panel) expressed as a percentage of TUNEL positive neurons out of the total number of TH-positive neurons. $N = 3$ each group. Bars and error bars represent the mean ± SEM. Statistical significance was determined using a one-way ANOVA followed by *post hoc* Tukey test for multiple groups. *$P < 0.05$; **$P < 0.01$.

D   Immunoblot of SN lysates (upper panel) and corresponding quantification (bottom panel). Quantification of band intensity (bottom). Mouse $N = 2$. $N = 3$ independent immunoblot analysis. Bars and error bars represent the mean ± SEM. Statistical significance was determined by an unpaired *t*-test. *$P < 0.05$; P**$< 0.01$.

E   Motor behavior tests at 6 weeks after treatment. $N = 8$ animals/group. Bars and error bars represent the mean ± SEM. Statistical significance was determined using a one-way ANOVA followed by *post hoc* Tukey test for multiple group comparison. *$P < 0.05$; N.S., not significant.

Source data are available online for this figure.

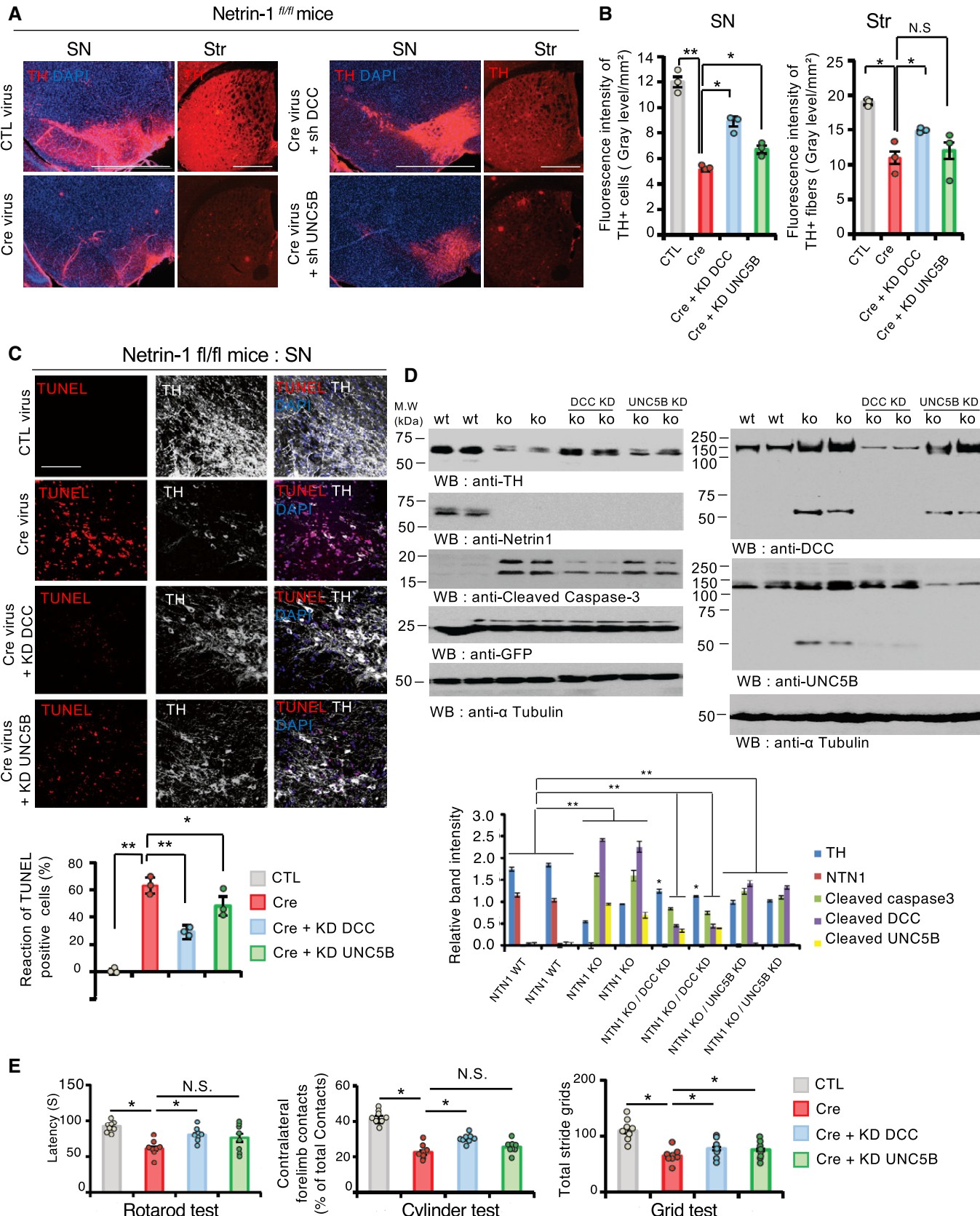

**Figure 3.**

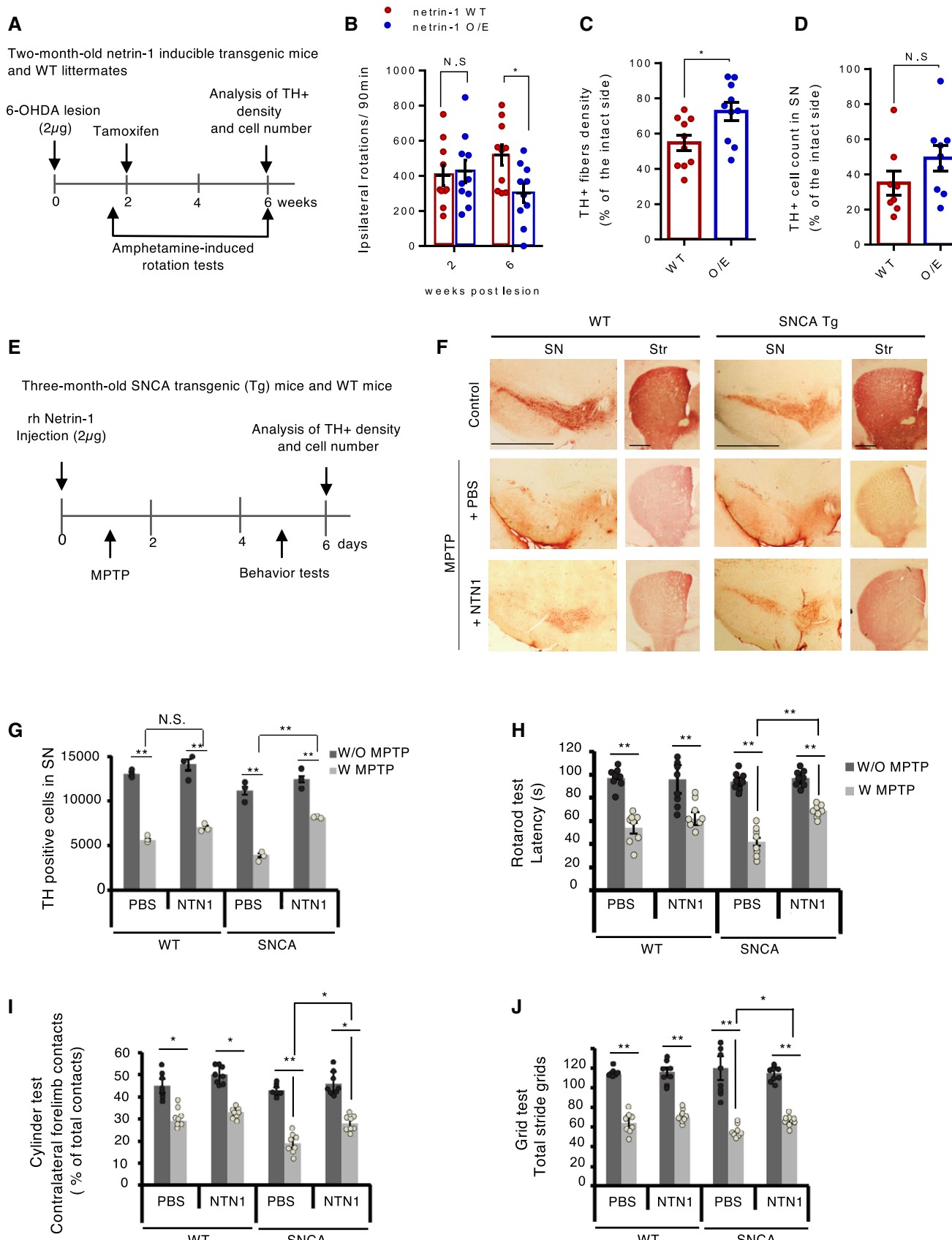

**Figure 4.**

**Figure 4.  Netrin-1 is neuroprotective in progressive and acute mouse models of PD.**

A   Experimental design in a unilateral 6OHDA mouse model of PD. Netrin-1 inducible mice and WT littermates were lesioned with 6-OHDA in two sites of the right striatum (2 × 1 µg). After 2 weeks, mice received i-p injections of tamoxifen to induce netrin-1 expression (O/E). Amphetamine-induced ipsilateral rotation behavior was assessed at week two and week six post-lesion then brains were taken to perform IHC analyses.

B   Amphetamine-induced rotations at two and six weeks after lesion. Individual values and Mean ± SD are shown, $n = 11$ animals in each group. Tukey *post hoc* analysis after two-way RM ANOVA, *$P < 0.05$ compared to WT.

C   Optical density of TH-positive fibers in the striatum. Individual values and mean ± SD, $n = 10$ animals, unpaired *t*-test, *$P < 0.05$.

D   TH-positive cell count in the SN. Individual values and mean ± SD, $n = 10$ animals, unpaired *t*-test.

E   Experimental design in the SNCA Tg MPTP acute mouse model of PD. WT and SNCA Tg mice were injected with netrin- 1 human recombinant protein in one site of the SN (2 µg). MPTP (18 mg/kg) was then injected (i.p.) every 2 h for a total of four doses over on 8 h period in 1 day.

F   TH staining by immunohistochemistry (Scale bar, 2,000 µm) in the SN (left) and striatum (right) of WT and SNCA Tg mice, 6 days after treatment.

G   TH-positive cell count bar graph. All experiments were repeated three times. We randomly selected two mice from each group for immunohistochemistry TH-positive cell analysis. Bars and error bars represent the mean ± SEM. Statistical significance was determined using a one-way ANOVA followed by *post hoc* Bonferroni test for multiple group comparison. **$P < 0.01$; N.S., not significant.

H–J   Motor behavioral assays (Rotarod rod test (H), Cylinder test (I), Grid test (J)) $N = 8$ animals each group. Bars and error bars represent the mean ± SEM. Statistical significance was determined using a one-way ANOVA followed by *post hoc* Bonferroni test for multiple group comparison. *$P < 0.05$; **$P < 0.01$.

Two weeks after, recombinant netrin-1 (10 µg) or GDNF as a positive control (10 µg) or vehicle (PBS 1X) were injected into those same three sites. To test the long-term effect of netrin-1 treatment, we monitored amphetamine-induced ipsilateral rotational behavior every two weeks for twelve weeks, and then, brains were collected for morphological analyses (Fig 5A). Whereas the ipsilateral rotational behavior of vehicle-treated rats significantly increased over time, rats treated with netrin-1 exhibited a stable rotational behavior (Fig 5B, left panel) that was significantly lower than rats treated with vehicle (Fig 5B, right panel). As a positive control, the injection of GDNF provided a similar neurorestorative effect than netrin-1 (Fig 5B). Immunohistochemical analyses showed that TH-positive striatal fiber density was significantly higher in the netrin-1 and GDNF-treated group than in the vehicle-treated group (Fig 5C) three months after 6-OHDA lesion, demonstrating that one single injection of netrin-1 provided stable neurorestorative effects in this model. The number of TH-positive cells in the SNpc was also higher, although not significantly, in the netrin-1-treated group compared with vehicle group (Fig 5D). To better characterize the neurorestorative effect of netrin-1 and compare it with the effect seen with GDNF, we performed distribution experiments using iodinated recombinant human netrin-1 ($^{125}$I-netrin-1) and iodinated recombinant human GDNF ($^{125}$I-GDNF). Contrary to GNDF, netrin-1 was not actively transported to the substantia nigra but was instead detectable in different regions of the striatum and in the frontal brain region comprising the cortex, 24 h after injection (Appendix Fig S4A and B). We also checked netrin-1(His-tagged) distribution by immunofluorescence 24 h after intrastriatal injection (Appendix Fig S4C) and found no signal in the SNpc, consistent with $^{125}$I-netrin-1 distribution experiments, but neither in the frontal part of the brain. This staining was restricted to the striatum (Appendix Fig S4C). Given their different distribution profile, these results suggest that netrin-1 may have a different mode of action than GDNF and may act locally on DA neuron axon terminals or other targets. Using primary culture of mouse ventral midbrain primary neurons, we showed that netrin-1 was not only able to promote the survival of dopamine neurons (Appendix Fig S4D) but also, as previously reported (Lin *et al*, 2005; Xu *et al*, 2010), elicited neurite outgrowth (Appendix Fig S4E). Thus, netrin-1 neurorestorative effect might not only result from a direct survival-promoting effect on dopamine neurons but also possibly on axonal growth and regeneration. Supporting this, we found that striatal levels of PTEN and phospho-S6 (PS6) ribosomal protein (Ser235/236), involved in the regulation of protein synthesis, were readily down and upregulated in response to netrin-1 intrastriatal injection in adult 6-OHDA rats, respectively (Appendix Fig S4F). We can also observe a rapid downregulation of DCC levels upon netrin-1 addition, which is in line with previous studies on neuronal guidance and growth reporting a fast decrease in DCC levels or cell presentation in response to netrin-1(Bai & Pfaff, 2011; DeGeer *et al*, 2013; Neuhaus-Follini & Bashaw, 2015).

**Figure 5.  Netrin-1 is neurorestorative in the rat 6-OHDA preclinical model of PD.**

A   Experimental design in a unilateral 6-OHDA rat model of PD. Rats were lesioned unilaterally with 6-OHDA into three sites of the right striatum (3 × 2 µg). After 2 weeks, rats received intrastriatal injections of recombinant human netrin-1, recombinant human GDNF or PBS (VEH). Amphetamine-induced behavior was assessed every 2 weeks for 12 weeks then brains were taken to perform IHC analyses.

B   Cumulative amphetamine-induced ipsilateral rotations. Progression of amphetamine-induced ipsilateral rotations over time (right panel). Mean + SD are shown, $n = 11–12$ animals in each group. Tukey *post hoc* analysis after two-way RM ANOVA, ****$P < 0.0001$ compared to the initial rotation score at week 2. Comparison of the treatment's effect between groups (left panel). Mean + SD are shown, $n = 11–12$ animals in each group. Tukey *post hoc* analysis after two-way RM ANOVA, *$P < 0.05$, **$P < 0.01$ compared to vehicle group.

C   TH-positive fibers or dopamine neurons fibers in the striatum of 6-OHDA-lesioned rats, ten weeks after treatment. Representative images of coronal sections of the striatum showing TH IHC staining (left panel), intact side on the left and 6-OHDA-lesioned side on the right. (Scale bar, 2,000 µm). Optical density measurement of TH-positive fibers in the lesioned side of the striatum (bar graph, right panel). Individual values and mean ± SD are shown, $N > 10$ animals in each group. Tukey *post hoc* analysis after one-way ANOVA, *$P < 0.05$, **$P < 0.01$ compared to vehicle group. Images are oriented with the dorsal part of the tissue at the top, and ventral part at the bottom

D   Dopamine cell bodies in the SN of 6-OHDA-lesioned rats, ten weeks after treatment. Representative images of coronal sections of the SN (lesioned side) showing TH-positive cell bodies stained by IHC (left panel) (Scale bar, 2,000 µm). TH-positive cell count in the SN (right panel). Individual values and mean + SD are shown, $N > 10$ animals in each group. Tukey *post hoc* analysis after one-way ANOVA, **$P < 0.01$ compared to vehicle group. Images are oriented with the dorsal part of the tissue at the top and ventral part at the bottom.

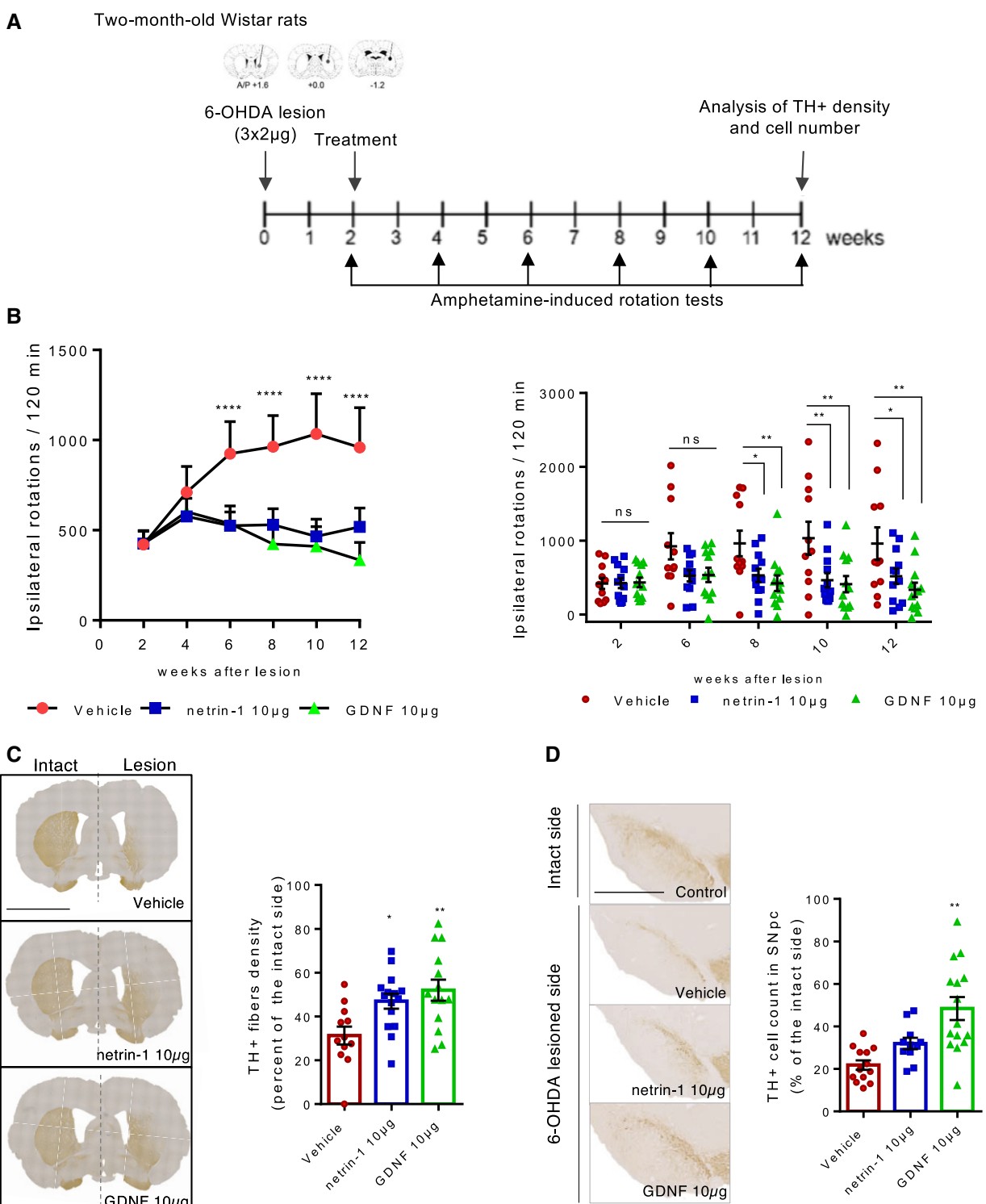

**Figure 5.**

## Netrin-1 is reduced in PD patient brain

To assess whether our findings in animals may be extended in the human pathology, we looked at netrin-1 status in PD patient brains. First, we compared *netrin-1* gene expression

levels in the SN of non-PD vs PD patients using a gene expression profiling dataset. We found that *netrin-1* expression was significantly decreased in the SN of PD patients compared to control patients (Fig 6A). Netrin-1 reduction in PD was confirmed at the protein level in nigral samples from PD patients

and age-matched control patients (Fig 5B). Moreover, the loss of netrin-1 in PD patients was accompanied by increased DCC and UNC5B levels and caspase cleavage (Fig 6B) as found in netrin-1 conditional KO mice (Fig 2F). Of note, the increased level of DCC/UNC5B was not seen using the above gene expression profiling dataset. Given our findings in mice showing that decreased levels of netrin-1 can trigger the loss of SN DA neurons, these results in human, even though at this step preliminary because of limited access to materials, may indicate that the reduction of netrin-1 in patients could contribute to the progression of PD.

# Discussion

### Netrin-1 as a unique regulator of adult dopamine neuron fate

The data presented here support a unique and critical role of netrin-1 on the maintenance of adult nigral neurons. Indeed, in the adult brain, netrin-1 is mainly expressed in the brainstem and notably in the substantia nigra. These structures are predominantly affected by Lewy pathology and neurodegeneration in PD. Especially, SNpc DA neurons are particularly susceptible to degeneration. Targeting features underlying DA neurons selective vulnerability might

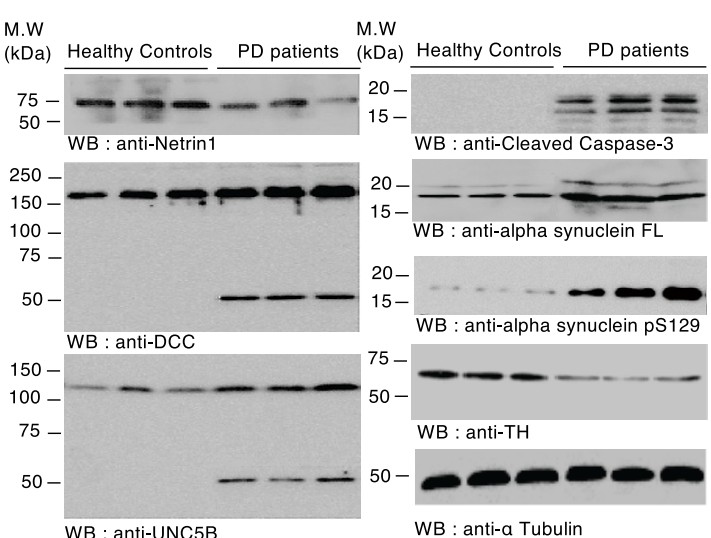

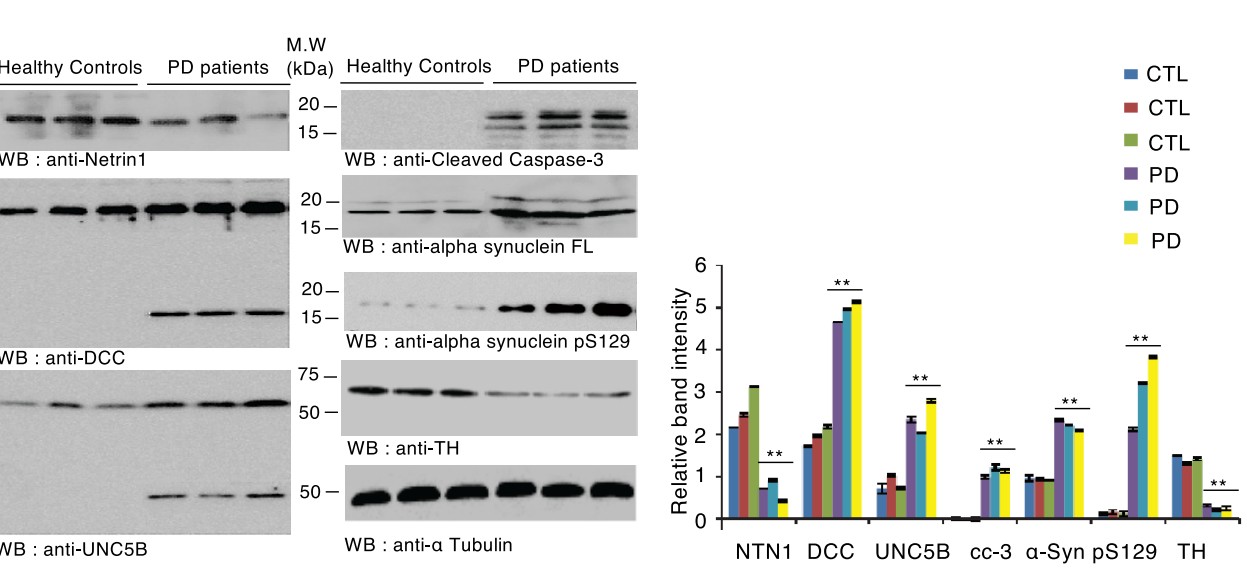

**Figure 6. Netrin-1 is reduced in the SN of PD patients.**

A   Netrin-1 gene expression profiling by array of substantiae nigrae from PD and non-PD (Control) patients using the GEO dataset GSE7621 that has a total of 25 samples (*n* = 9 control and *n* = 16 PD cases). Unpaired *t*-test, ***P* < 0.0005, Mean ± SD are shown.

B   Immunoblot of netrin-1, DCC, UNC5B, active caspase-3, alpha-synuclein FL, alpha-synuclein pS129, and tyrosine hydroxylase (TH) levels in age-matched controls (*n* = 3) vs Parkinson's disease patients brain samples (*n* = 3) (left panel). Band quantification bar graph (right panel). Bars and error bars represent the mean ± SEM. Statistical significance was determined using a one-way ANOVA followed by *post hoc* Tukey test for multiple group comparison. ***P* < 0.01. *N* = 3 replicates.

Source data are available online for this figure.

represent a therapeutic approach to slow or halt neurodegeneration. Studies reported, both in human and mouse midbrain, that ventral SNpc DA neurons exhibited a stronger signal for DCC than other midbrain DA neurons (from dorsal SNpc and the VTA; Osborne *et al*, 2005; Reyes *et al*, 2013), suggesting that DCC protein level may be a marker for DA neurons that are most vulnerable to degeneration. However, it was unclear whether the expression pattern of DCC was causally linked to the selective DA neuron vulnerability, and how DCC may modulate the susceptibility of DA neurons to degeneration. We showed that silencing endogenous netrin-1 in the adult mice substantia nigra resulted in a dramatic (70%) loss of dopamine neurons. Of note, there is currently no reported growth factor, including GDNF, which deletion induces such a massive loss of adult dopamine neurons, which further stresses the key role of endogenous netrin-1 on the maintenance of these neurons. Additionally, simultaneous deletion of DCC partly prevented dopamine neuron loss, suggesting that DCC, via its death-inducing activity, regulates DA neurons survival. However, it is fair to say that we cannot exclude that not only the death activity of DCC is important to mediate this DA loss but also alternative loss of netrin-1-induced DCC signaling. Thus, netrin-1 receptor DCC might be a double-edged sword for SNpc DA neurons depending on netrin-1 availability: promoting survival (and outgrowth) in healthy conditions and cell death in conditions of low trophic support. As SNpc DA neurons are thought to be prone to oxidative stress and inflammation due to their unique morphology and pacemaker activity, DCC dual signaling could be a safeguarding mechanism, eliminating damaged or "diseased" neurons and protecting healthy neurons from additional harm. Altogether, netrin-1 appears to be a unique protein able to maintain adult dopamine neurons. Further studies on the role of endogenous netrin-1 in adult SNpc DA neurons and how netrin-1 is decreased in PD might bring new understandings on SNpc DA neuron selective vulnerability and PD pathogenesis.

### Toward recombinant netrin-1 injection in PD

Beside dopamine neuron loss, the other key hallmark of PD is Lewy pathology. Although a large part of our study was focused on the first hallmark, we also showed that netrin-1 silencing was associated with increased alpha-synuclein levels. Moreover, we observed that there was a decrease of netrin-1 expression and protein levels in an age-dependent manner in the adult substantia nigra of SNCA transgenic mice, suggesting that there might be a link between alpha-synuclein and netrin-1 levels. Importantly, combining different PD animal models, we show for the first time that injection of recombinant netrin-1 was not only able to protect neurons against alpha-synuclein toxicity, *in vivo*, but could also restore dopamine axonal projections in the striatum.

Even though further preclinical work will be required, the present work sheds light on the therapeutic relevance of targeting netrin-1 signaling in PD. We demonstrated *in vivo*, using netrin-1 conditional knockout mice, that netrin-1 reduction could trigger dopamine neuron demise and motor deficits, in part through DCC-induced cell death, and thus, netrin-1 loss could contribute to PD development or progression. Inversely, both administration of netrin-1 recombinant protein in the MPTP/SNCA mice model and in 6-OHDA-lesioned rats showed significant neuroprotective and neurorestorative effect on mature nigral dopamine neurons. Thus,

local injection of full-length recombinant netrin-1 or administration of netrin-1 mimetic small molecules might thus represent a promising disease-modifying treatment for PD. Indeed, brain injection of other recombinant trophic factors have recently been assessed in PD human trials (Whone *et al*, 2019) and we have shown here that a single injection provided long-term benefit (i.e., we assessed repeated netrin-1 injection without significant improvement compared to single injection) in the 6-OHDA-lesioned rats, thus, supporting the view that a therapeutic strategy aiming at delivering recombinant netrin-1 in the brain of PD patient should be assessed.

## Materials and Methods

### Human tissue samples

Post-mortem brain samples were dissected from frozen brains of PD patients and aged-match non-demented controls from the Emory Alzheimer's Disease Research Center (ADRC). The study was approved by the Biospecimen Committee (The Goizueta ADRC, Emory University). PD was diagnosed according to the criteria of the Consortium to Establish a Registry for PD and the National Institute on Aging (Morris *et al*, 1989; Mirra *et al*, 1991). Informed consent was obtained in all cases.

### Animals

All animals were housed in filter-topped cages under a 12 h light/dark cycle at an ambient temperature of 22°C. Females and males were kept separately. Tap water and rodent chow were available ad libitum.

Investigators were blinded to the group allocation during the animal experiments.

### Netrin-1$^{+/LacZ}$ generation and salmon gal staining

Netrin-1$^{+/lacZ}$ mice were generated as described (Serafini *et al*, 1996). P60 mice were euthanized (pentobarbital 90 mg/kg, ip) and fixed (formaldehyde 4%) through perfusion. Brains were removed, post-fixed for 4 h, and embedded in paraffin. Coronal sections of 10 μm thickness were stored at −80°C until salmon gal staining.

For salmon gal staining, brain cryosections were incubated overnight in pre-stain solution without substrate (potassium ferrocyanide 200 mM, MgCl$_2$ 4 mM, NP40 0.04% in PBS) at 37°C to reduce endogenous β-galactosidase activity. Cryosections were then washed three times in PBS and incubated for 1 h at 37°C in the staining solution (1 mg/mL Salmon-Gal, 0.33 mg/mL NBT, MgCl$_2$ 2 mM, NP40 0.04% in PBS) then rinsed in PBS.

### Netrin-1$^{fl/fl}$ conditional KO mice

Netrin-1$^{flox(fl)}$ mice (C57BL/6 background) were obtained from the Jackson Laboratory (Stock No:028038). These mutant mice possess loxP sites flanking the first exon of the *Ntn1* gene. Homozygous mice are viable and fertile. Removal of the floxed sequence by cre recombinase generates a null allele resulting in the silencing of netrin-1 in the cre-targeted tissue. The generation of the Netrin-1$^{fl}$ allele and Netrin-1$^{fl}$ mice are described in Bin *et al* (2015) and

available on Jackson Laboratory website. Netrin-1 knock out in the SN was achieved injecting stereotaxically AAV6-Cre-eGFP virus vector (vs AVV6-GFP Control virus) into the right SN of three-month-old Netrin-1$^{fl/fl}$ mice. Mice were randomly allocated to experimental groups.

Animal care and handing were performed according to the Declaration of Helsinki and Emory Medical School guidelines. The protocol was reviewed and approved by Emory Institutional Animal Care and Use Committee.

## Inducible netrin-1 transgenic mice

To generate breeder pairs of control (WT) and netrin-1 conditional overexpressing (O/E) mice, we crossed mice bearing a tamoxifen-dependent Cre recombinase allele (*CAG-CreERT2*$^{+/+}$) with a transgenic mouse line containing the human netrin-1 (NTN1) cDNA preceded by a lox-stop-lox cassette (LSL) inserted in the Rosa26 locus as described in Lahlali *et al* (2016). The litter carries one copy of the CreERT2 transgene and was genotyped for LSL–NTN1. Two-month-old mice (*CAG:CreERT2*$^{+/-}$;*Rosa26-LSL*-NTN1$^{+/+}$) and WT littermates (*CAG:CreERT2*$^{+/-}$;*Rosa26-LSL*-NTN1$^{-/-}$) were used for the experiments and received intrastriatal injection of 6-OHDA. Two weeks after, mice received three intraperitoneal (i-p) injections (one injection very other day) of 200 µl fresh tamoxifen solution (Sigma-Aldrich, T5648) to induce netrin-1 O/E. Tamoxifen solution for injection was dissolved in corn oil (Sigma-Aldrich, C8267) at a concentration of 10 mg/ml. Experimental procedures were approved by the Committee for Animal Experiments of the Rhône-Alpes Region (CE015), France (authorization no.: APAFIS#1565).

## Acute MPTP intoxication in WT and SNCA transgenic mice

C57BL/6 WT mice and human SNCA transgenic C57Bl/6 mice carrying a transgene containing human *SNCA* gene and a knockout allele of the mouse *Snca* gene were obtained from Jackson Laboratory (Stock No: 023837).

C57BL/6 WT and SNCA Tg mice (age 8–12 weeks, 8–10 per group) received i.p. injections of vehicle or 1-Methyl-4-phenyl-1,2,3,6-tetrahydropyridine (MPTP) at indicated doses (free base in PBS; Sigma) every 2 h for a total of four doses over an 8 h period in 1 day (18 mg/kg). Behavior tests were performed 5 days after the last dose of MPTP.

Animal care and handing were performed according to the Declaration of Helsinki and Emory Medical School guidelines. The protocol was reviewed and approved by Emory Institutional Animal Care and Use Committee.

## Unilateral 6-OHDA rat model of PD

Six-week-old male Wistar rats were purchased from Janvier Labs (Genest Saint-Isle, France). After one-week acclimatization followed by a one-week habituation period, rats (two-month-old, 250 g body weight) were lesioned intrastriatally and unilaterally with 6-OHDA. Rats were divided into treatment groups according their amphetamine-induced rotations at 2 weeks post-lesion. Motor behavior (ipsilateral rotation tests) was assessed every two weeks for twelve weeks as from 6-OHDA lesioning, and then, animals were euthanized to perform immunohistochemical analyses.

Experimental procedures were approved by the Committee for Animal Experiments of the Rhône-Alpes Region (CE015), France (authorization no.: APAFIS#1565).

## Stereotaxic injection and treatments

Stereotaxic coordinates were determined according to the rat brain atlas of Paxinos and Watson (1996) and the mouse brain atlas of Franklin and Paxinos (2001).

### AAV6 virus and lentiviral-ShRNA injections in Netrin-1$^{fl/fl}$ mice

The AAV6-Cre-eGFP virus (with Capsid from AAV6 and ITR from AAV2) was used to KO *Netrin-1* expression in Netrin-1$^{fl/fl}$ mice. This AAV serotype 6 virus vector expresses Cre Recombinase (tagged with nuclear localization sequence) and eGFP marker, driven by CMV promoter. AAV6-eGFP virus vector (with Capsid from AAV6 and ITR from AAV2) expressing eGFP under the control of the promoter CMV was used as control virus. AAV6-Cre-eGFP and AAV6-eGFP viruses vectors were purchased from Vector Biolabs (Cat. No: 7019 and 7008, respectively).

Three-month-old Netrin-1$^{fl/fl}$ mice were anesthetized with isoflurane. Meloxicam (5 mg/kg) was injected subcutaneously for analgesia (Loxicom, Norbrook, USA). Unilateral stereotaxic injection of AVV6-Cre-eGFP or control AAV6-eGFP viruses vectors and lentiviral-ShRNA silencing *Dcc* or *Unc5b* was performed at coordinates corresponding to the substantia nigra: anteroposterior (A/P): ±3.1 mm, mediolateral (M/L): ±1.2 mm from Bregma, and dorsoventral (D/V): 4.3 mm from dura surface. Each site received 2 µl of viral constructions at a rate of 0.25 µl/min, using 10 µl Hamilton syringe. Viral titer was $1 \times 10^{12}$ vg/ml for AAV6 virus constructions and $1 \times 10^{10}$ vg/ml for lentiviral constructions. The needle was kept in place for 5 min after the injection was completed, then gently removed. Mice were placed on a heating pad until they recovered from the anesthesia.

### Recombinant protein injections in 6-OHDA-lesioned animals

All stereotaxic and surgical procedures were performed under isoflurane anesthesia, as described in (Voutilainen *et al*, 2009). Meloxicam (2 mg/kg) was used for analgesia and lidocaine–adrenaline solution (10 mg/ml lidocaine, 0.005 mg/ml adrenaline) for local anesthesia before surgery. 6-OHDA (Sigma-Aldrich, H4381) was injected at the rate of 0.5 µl/min into the right striatum using a 33G, 10 µl, outer diameter 0.21-mm blunt needle (NanoFil, World Precision Instruments, NF33BL-2).

In mice, 2 µg of 6-OHDA (2 × 1 µg; 1 µl/site of 1 µg/µl solution) were injected into two sites of the dorsal striatum (coordinates relative to Bregma and dura: A/P: +1, M/L: +2.2, D/V: −3 mm and A/P:0, M/L: +2.2, D/V: −3 mm).

In rats, 6 µg of 6-OHDA (3 × 2 µg; 1.5 µl/site of a 1.33 µg/µl solution) was injected into three sites of the striatum (A/P: +1.6 mm, M/L: +2.8 mm, D/V: −5.5 mm ; A/P : 0 mm, M/L: +4.1 mm, D/V: −6 mm and A/P: −1.2 mm, M/L: 4.5 mm, D/V: −6 mm from Bregma). The dose of 6-OHDA was calculated as a free base and dissolved in degassed saline with 0.1 % sodium metabisulfite to prevent oxidation. After each injection, the needle was kept in place for 4 min to minimize backflow of the solution. For neurorestorative experiments in rats, 10 µg (1.5 µl/site of a 2.22 µg/µl solution) of recombinant human netrin-1 (R&D, 6419-N1-025/CF) or 10 µg (1.5 µl/site) of

recombinant human GDNF (Eurobio, PCYT-305) or 1.5 μl/site of PBS Vehicle (Gibco) were injected two weeks after 6-OHDA lesioning, into the same three sites of injection. For distribution studies, 10 μg (1.5 μl/site) of iodinated $^{125}$I-netrin-1 (R&D, 6419-N1-025/CF) or $^{125}$I-GDNF was injected similarly.

## Behavioral tests

### Netrin-1$^{fl/fl}$ mice

Loss of motor function was tested 6 weeks following virus injection. Behavioral tests included the rotarod test, cylinder test, and grid tests. Rotarod test: Mice were trained for 3 days using the Rotarod (San Diego Instruments) at a slow rotational speed (5 rpm) for a maximum of 10 min. The rotational speed and latency to fall from the accelerating Rotarod were recorded after three test trials on the test day. The rotational speed of Rotarod was modulated from 0 rpm to a maximum 40 rpm. It was gradually increased during the trial at a rate of 0.1 rpm/s. Cylinder test: Mice were placed individually into a glass cylinder (12 cm diameter × 22 cm height) and were recorded with video camera. The recorded files were analyzed by a blinded observer. Between 20 and 30 wall touches per animal (contacts with fully extended digits executed with the forelimb ipsilateral and contralateral to the lesion) were counted. Grid test: the grid was inverted, so the mice were hanging upside down by their paws. Animals were recorded for up to 60 s, and then, 10 s-segments scored for the number of grids crossed at each step.

### Amphetamine-induced ipsilateral rotations in 6-OHDA-lesioned rats and mice

Amphetamine-induced rotational activity was monitored with the Rotameter system from Omnitech electronics, Inc, USA comprising of a cylindrical test chamber (30 cm × 30 cm), rat or mice harnesses and sensors for circling for data collection. Animals were placed in the test chamber and, after a habituation period of 15 min, a single dose of d-amphetamine (Tocris; CAS-51-63-8) was injected i.p. The number of full (360°) clockwise and counterclockwise turns was recorded for a period of 120 min for rats and 90 min for mice. Net ipsilateral turns were automatically calculated by the Fusion software (Omnitech electronics) subtracting the turns to the left from the turns to the right side.

In rats, rotation tests were done 2–3 days before, and 2, 4, 6, 8, and 10 weeks after treatment (netrin-1, GDNF, or vehicle injection) upon d-amphetamine (2.5 mg/kg). Results are shown as the total (cumulative) of net ipsilateral turns performed over 120 min.

In mice, rotation tests were done 2 and 6 weeks after 6-OHDA lesion, upon d-amphetamine (5 mg/kg). Results are shown as the total of net ipsilateral turns performed over 90 min.

Rats and mice that did less than 100 ipsilateral turns at the first rotation test, 2 weeks after 6-OHDA lesion, were excluded from the experimental procedure. Indeed, according to Bäck and colleagues (Bäck et al, 2013), rats exhibiting less than 100 ipsilateral amphetamine-induced rotations /120 min at 2 weeks after 6-OHDA lesion can be considered as not successfully lesioned. We decided to apply the same threshold for mice.

## Perfusion and tissue processing for immunofluorescence and immunochemistry

Mice and rats were anesthetized with an overdose of sodium pentobarbital (90 mg/kg, i.p.) and perfused intracardially with PBS followed by 4% paraformaldehyde in a 0.1 M sodium phosphate buffer, pH 7.4. Brains were removed, post-fixed overnight, and stored in sodium phosphate buffer containing 30% sucrose at 4°C. Serial coronal frozen sections of 40 μm (rat samples) or 30 μm (mouse samples) thickness were made using a cryostat. Six sets of sections were collected in a cryoprotectant solution (0.1 M phosphate buffer, pH 7.4, 20% glycerol and 2% dimethyl sulfoxide) and stored at −20°C until immunohistochemical or immunofluorescence processing.

## Key resources Table

| Antibodies | Dilution ratio | Source | Manufacturers | Article number |
|---|---|---|---|---|
| α-synuclein FL | 1:1,000 | Mouse | Santa Cruz | SC-69977 |
| α-synuclein-pS129 | 1:1,000 | Rabbit | LS Bio | LS-C380861-1 |
| Cleaved caspase 3 | 1:500 | Rabbit | Cell signaling | #9661 |
| DAT | 1:500 | Rabbit | Abcam | Ab184451 |
| DCC | 1:500 | Mouse | Santa Cruz | SC-515834 |
| His-tag | 1:500 | Mouse | Abcam | Ab18184 |
| Netrin-1 | 1:1,000 | Rabbit | Santa Cruz | SC-20786 |
| Netrin-1 | 1:500 | Mouse | Santa Cruz | SC-293197 |
| Netrin-1 | 1/500 | Rabbit | Abcam | Ab126729 |
| TH | 1:750 | Mouse | Santa Cruz | SC-25269 |
| TH | 1:750 | Rabbit | Abcam | Ab112 |
| UNC5B | 1:500 | Mouse | R&D systems | MAB1006 |
| VMAT2 | 1:500 | Rabbit | Novus | NBP1-69750 |

## Immunofluorescence

Free-floating slices were rinsed in PBS then permeabilized and blocked with PBS-BT (50 mM Tris–HCl, 150 mM NaCl, 3% bovine serum albumin (BSA), 0.1% Triton X-100, pH 7.4) blocking solution for 1 h. Afterward, the sections were incubated with primary antibodies (see key resource table) in a 2% normal donkey serum (NDS) and 0.3% Triton X-100 PBS solution on a shaker overnight at 4°C. The next day, sections were rinsed and incubated with corresponding secondary antibodies directly conjugated with fluorophores (1:500, Cy3, Cy5, and Alexa Fluor 488 conjugate from Jackson ImmunoResearch) for 2 h at room temperature. Finally, slices were rinsed in PBS and mounted (Sigma-Aldrich, F4680). For DAT and netrin-1 immunostaining, antigen retrieval was performed in citrate buffer (pH 9) at 80°C.

## Immunohistochemistry

Free-floating sections were processed for tyrosine hydroxylase (TH) immunohistochemistry. Endogenous peroxidase activity was quenched for 5 min in a 3% H$_2$O$_2$, 10% methanol, PBS-1X (PBS) solution and then washed with PBS. Sections were next

preincubated with 2% normal donkey serum (Jackson ImmunoResearch, 017-000-121), 0.3% Triton X-100 in PBS to block non-specific staining. Then, sections were incubated overnight at room temperature with a 1:500 dilution of chicken anti-tyrosine hydroxylase antibody, rinsed, and incubated 2 h with a 1:300 dilution of biotinylated donkey anti-chicken antibody (Jackson ImmunoResearch, 703-065-155) before 1 h incubation with the avidin–biotin peroxidase complex using the Elite ABC Vectastain kit (Vector Laboratories). Then, TH staining was revealed using 3,3'-diaminobenzidine (DAB) as a chromogen. Sections were then put onto glass microscope slides, dried overnight at room temperature, dehydrated, and mounted in a xylene-containing medium.

## Estimation of striatal TH-positive fiber optical density

The optical density of the striatal TH-positive fibers were determined from three coronal striatal sections as described in (Voutilainen et al, 2009). Every sixth section between A/P: +1.6 mm and A/P: −0.20 mm for rats and between A/P: +1 mm and A/P: 0 mm for mice was cut on a cryostat and processed for TH immunohistochemistry. Sections were scanned with an automated scanner (Axio Scan.Z1, service provided by the Ciqle imaging platform, Lyon). The images were converted to 16-bit gray scale, and the optical density of striatal fibers from control and lesioned side were measured using the integrated densities divided by area in ImageJ (NIH). Only striatal parts corresponding to the dorsal striatum, i-e, caudate nucleus and putamen (CPu) were comprised in the region of interest (ROI) selected for the analysis. Nucleus accumbens (core and shell) was not comprised in the ROI. The corpus callosum which is devoided of TH signal was used as an internal control for nonspecific background staining. Brain structures were identified and selected according to the rat brain atlas of Paxinos and Watson (1996) and the mouse brain atlas of Franklin and Paxinos (2001). Data are presented as percentage of the intact side.

## TH-positive cell counting

Unbiased stereological cell counting was performed using the Stereo Investigator software (MBF Bioscience, Williston, VT) (Voutilainen et al, 2009). All substantia nigra sections were stained with TH antibody. They were analyzed under the randomly placed counting frames (50 × 50 μm) on a counting grid (120 × 120 μm). Optical dissector of 22 μm with 2 μm upper and lower guard zones was used. The boundaries of substantia nigra were outlined under magnification of the 5× objective lens, and cells were counted with the 40× objective lens using an Olympus BX53 microscope. The total number of neurons in the substantia nigra was estimated using the optical fractionator method.

Alternatively, the number of TH-immunoreactive cells in the SNpc was determined by using a deep Convolutional Neural Network method using Aiforia platform as described in (Penttinen et al, 2018) by a blinded observer. Images taken with whole slide scanner (Pannoramic P250 Flash II whole slide scanner 3DHistech, Budapest, Hungary) with extended focus at a resolution of 0.22 μm/pixel from six adjacent 30 μm-thick nigral sections. Images were uploaded to Aiforia™ image processing and management platform (Fimmic Oy, Helsinki, Finland) and analyzed using a deep CNN algorithm and supervised learning. Data are presented as a percentage of the intact side.

## Immunoblotting

Cells and brain samples were lysed and, if necessary, homogenized in lysis buffer (50 mM Tris, pH 7.4, 40 mM NaCl, 1 mM EDTA, 0.5% Triton X-100, 1.5 mM $Na_3VO_4$, 50 mM NaF, 10 mM sodium pyrophosphate and 10 mM sodium β-glycerophosphate, supplemented with a cocktail of protease inhibitors). Lysates were then centrifuged for 20 min at 4°C 15,000 rpm and protein concentration of the supernatant was measured with the Pierce BCA Protein Assay Kit (Part No. 23225). The supernatant was denatured at 95°C in Laemmli buffer. After loading and running proteins in a SDS–PAGE gel, the samples were transferred to a nitrocellulose membrane. Membrane blocking and antibody staining were performed according to the primary antibody manufacturer's instructions. Primary antibodies to the following targets were used: alpha-synuclein FL (Santa Cruz, Cat# SC69977) alpha-synuclein pS129 (LS bio, Cat# LS-C380861-1); Tyrosine Hydroxylase (Santa Cruz, SC-25269; Abcam, Cat# ab112); Netrin-1(Santa Cruz, Cat# SC20786 or SC-293197; Abcam, Cat# ab126729); UNC5B (R&D systems, Cat#MAB1006); DCC (Santa Cruz, Cat# SC515834); and cleaved caspase-3 (Cell signaling Cat# 9661).

## TUNEL assay

Dopamine neuron death was detected with an *in situ* cell death detection kit TMR Red (Roche, Cat# 12156792910). The apoptotic index was expressed as the percentage of TUNEL-positive neurons out of the total number of TH-positive neurons.

## Preparation and use of [125]I-labeled recombinant protein

Recombinant human netrin-1 (R&D) and GDNF (PeproTech) were iodinated with [125]I-Na using the lactoperoxidase method. Proteins were dissolved in 30 μl of 0.25 M phosphate buffer, pH 7.5, and mixed with [125]I-Na (1 mCi 37 mBq; GE Healthcare). The reaction was started by adding lactoperoxidase 10 μl of 50 μg/ml and 0.05% $H_2O_2$. The mixture was incubated at room temperature for 20 min and the reaction was stopped by adding 3 vol of 0.1 M phosphate buffer, pH 7.5, containing 0.1 M NaI, 0.42 M NaCl, and 25 μl of 2.5% BSA. Free iodine and iodinated growth factors were separated by Sephadex G-25 columns (PD10; GE Healthcare). For column equilibrium and elution, 0.1 M phosphate buffer, pH 7.5, with 1% BSA was used. The iodinated proteins were concentrated by using YM-10 Centricon columns (Millipore). The specific activity of [125]I-labeled netrin-1 and GDNF was > $10^8$ cpm/μg protein.

## Gamma counting and autoradiographic analyses

Portions of brain tissue were used for gamma counting on Wallac Wizard 1480 gamma counter, and the remaining were sectioned for autoradiography.

### *Autoradiographic analysis of the distribution of [125]I-proteins*
Rats that received intrastriatal injection of [125]I-netrin-1 or [125]I-GDNF were perfused 24 h after stereotaxic injections. Coronal paraffin sections (7 μm thick) were juxtaposed against an autoradiography film (Kodak BioMax MS) for 4 weeks.

### Quantification of $^{125}$I-proteins in brain tissues after intrastriatal injections

The amount of intrastriatally administered proteins in different brain structures was determined after perfusions. The brain was removed from the skull, and the hippocampus, substantia nigra, striatum, and cortex were dissected out, and the wet tissue was weighed. Results are expressed as counts per minute per milligram of wet weight.

### Primary cortical neuron culture

Rat cortical primary neurons were cultured as previously described (Kang *et al*, 2018). At DIV 9, primary neurons were treated with netrin-1 human recombinant protein (100 and 200 ng/ml), netrin-1 blocking antibody (2F5; 0.5 and 5 μg), or interfering DCC ECD peptide (4Fbn; 0.5 and 5 μg) for 24 h, and then, the LDH levels were measured in the medium and cell lysates collected for Western blot analyses.

### Ventral midbrain primary culture

Dopamine primary neurons were prepared from the ventral midbrain of embryonic mice on day 13 of gestation. Ventral midbrain was isolated and dissected in ice-cold Dulbecco's medium + 0.2% BSA. Fragments were dissected into tiny pieces and collected in 2 ml vials. Then, pieces were washed three times in HBSS ($Ca^{2+}$ and $Mg^{2+}$ free) medium. Pieces were dissociated in HBSS containing 0.01% trypsin for 20 min at 37°C. Cells were dissociated by soft trituration in FBS medium containing 1 μg/ml of DNase I. After trituration, cells were washed in DA neuron culture medium (DMEM F12, N2 1X, 0.36% D-(+)-Glucose (wt/vol), primocin 100 μg/ml) and plated on 96-well plates coated with poly-L-ornithine at a density of $5 \times 10^4$ cells/well and let to adhere for 1 h. Then, cells were treated with netrin-1 or GDNF. The treatment was renewed every 2 days for 4 days.

After treatment, primary neurons were fixed with 4% formaldehyde for 15 min and permeabilized with PBS solution containing and 0.2% Triton X-100 for 20 min. Then, rinsed and incubated 1 h with a PBS blocking solution containing 2% NDS. Cells were next incubated with TH (Abcam, primary antibody ab76442) primary antibody overnight at 4°C. The next day, cells were rinsed and incubated for 1 h with donkey anti-chicken fluorescent secondary antibody and then rinsed and kept at 4°C in PBS.

### Netrin-1 gene expression profiling dataset

Netrin-1 gene expression profiling in the SN of PD patients was done using a gene dataset available on the gene expression omnibus (GEO) repository (GDS2821 accession number). We selected this dataset because the corresponding study (Papapetropoulos *et al*, 2006) included a large number of participants, nine control patients and 16 PD patients, and very strict RNA quality control criteria were used. Gene expression profiling was done using Affymetrix Human Genome U133 Plus 2.0 GeneChip arrays. We analyzed probe set data for NTN1 in the substantia nigra and selected NTN1 probe set on the criterion of "present" (detectable) call.

### Quantification and statistical analysis

When value represents a different individual, individual values are shown, and data are expressed as mean $\pm$ SD to show the variation among values. When means are compared, data are expressed as mean $\pm$ SEM, these data are obtained from three or more independent experiments. Representative morphological images obtained from at least three experiments with similar results were provided. ImageJ 1.47 software was used to analyze IHC and IF experiments, and Image Lab™ software for Western blots analysis. The statistical analysis of results was performed using GraphPad6 (Prism) software. All data were tested for normal distribution in order to analyze results accordingly using parametric or non-parametric tests. To compare results between two groups, Student's unpaired *t*-test was used. When more than two groups were compared, one-way ANOVA followed by Tukey *post hoc* test was applied. For repeated measures, a repeated-measures (RM) ANOVA or 2-way ANOVA test was performed followed by Tukey multiple comparisons *post hoc* test. All tests were two-sided. Assessments with $P < 0.05$ were considered significant.

## Data availability

This study includes no data deposited in external repositories.

**Expanded View** for this article is available online.

## Acknowledgements

We thank ADRC at Emory University for human PD patients and healthy control samples. This work was supported by grants from NIH RF1 (AG051538) to K. Y., and a grant from National Natural Science Foundation (NSFC) of China (No. 81528007) to K.Y, by grants from ANR to PM and MJF Foundation (Grant #13945) to PM. MJ was supported by fellowships from LabEX DEVweCAN, the ERC, France Parkinson association. MS was supported by the Jane and Aatos Erkko Foundation and by the Academy of Finland. We thank N Rama, M Airavaara, and K Albert for excellent helpful suggestions and technical assistance.

## Author contributions

MJ, MHV, EHA, MS, KY, and PM developed rationale and designed the experiments. MJ, EHA, and MHV performed most of the experiments and data analysis. EHA and SSK performed studies on netrin-1$^{fl/fl}$ and SNCA tg mice experiments. MJ performed studies on netrin-1 inducible transgenic mice with JF assistance. MJ and MHV performed rat experiments. CG provided technical assistance. Neuron counting using Aiforia platform was performed by TV. Mesencephalic primary culture experiments and analysis were performed by LY. Writing and editing by MJ, MHV, KY, MS, and PM.

## Conflict of interest

The authors declare that they have no conflict of interest.

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
