## [Review Process File · The EMBO Journal]

Netrin-1 and its receptor DCC modulate survival and death of dopamine neurons and Parkinson's disease features

Melissa Jasmin, Eun Hee Ahn, Merja Voutilainen, Joanna Fombonne, Catherine Guix, Tuulikki Viljakainen, Seong Su Kang, Li-ying Yu, Mart Saarma, Patrick Mehlen, and Keqiang Ye
DOI: [10.15252/embj.2020105537](https://doi.org/10.15252/embj.2020105537)

Corresponding author(s): Patrick Mehlen (mehlen@lyon.fnclcc.fr) , Keqiang Ye (kye@emory.edu)

Review Timeline:

Submission Date:	6th May 20
Editorial Decision:	24th Jul 20
Revision Received:	1st Sep 20
Editorial Decision:	22nd Oct 20
Revision Received:	4th Nov 20
Accepted:	12th Nov 20

Editor: Karin Dumstrei

Transaction Report:

Dear Patrick,

Thank you for submitting your manuscript to The EMBO Journal. I am very sorry for the delay in getting back to you with a decision, but I was waiting for the input from a third referee who has unfortunately still not returned the report and so I will go with the two reports that I have on hand.

As you can see from the comments there is an interest in the analysis. However, both referees also indicate significant revisions are needed to consider publication here. Should you be able to address the raised concerns with the inclusion of additional data then I would consider a revised version. I should point out that I need the support from both referees to move forward with the manuscript for publication here.

I am happy to discuss the raised points further and maybe it would be most helpful to do so via phone or video.

When preparing your letter of response to the referees' comments, please bear in mind that this will form part of the Review Process File, and will therefore be available online to the community. For more details on our Transparent Editorial Process, please visit our website:

<https://www.embopress.org/page/journal/14602075/authorguide#transparentprocess>

I thank you for the opportunity to consider your work for publication. I look forward to discussing your revisions further.

Yours sincerely,

Karin Dumstrei, PhD
Senior Editor
The EMBO Journal

- a point-by-point response to the referees' comments, with a detailed description of the changes made (as a word file).
- a word file of the manuscript text.

- individual production quality figure files (one file per figure)
 - a complete author checklist, which you can download from our author guidelines (<https://www.embopress.org/page/journal/14602075/authorguide>).
 - Expanded View files (replacing Supplementary Information)
- Please see out instructions to authors
<https://www.embopress.org/page/journal/14602075/authorguide#expandedview>

The revision must be submitted online within 90 days; please click on the link below to submit the revision online before 22nd Oct 2020.

Referee #1:

In this paper, Jasmin et al. al show that silencing netrin-1 in the adult SN leads to a loss of dopamine neurons resulting in motor deficits in mice. Then, they examine the role of netrin-1 in different animal models of PD. They show that overexpression of netrin-1 or administration of netrin-1 is neuroprotective in mouse and rat models. They have also observed a reduction of netrin-1 expression in PD patient brain samples.

Overall, this is a very interesting study that uncovers novel functions for netrin-1 and its receptor DCC in adult dopamine neurons, suggesting a potential therapeutic role of these proteins in PD treatment. In general, the data are convincing, well performed with appropriate controls. The quantification of the data and statistics are properly done in the majority of the figures. However, I disagree with some important conclusions of the paper and will explain my reasons below. In addition, there are some issues that need to be clarified, as detailed in the comments below:

Comments:

- 1) Figure 1: it would be nice to show a detailed staining of DCC and UNC5B in the SN regions as done for netrin-1 in Figure 1B.
- 2) Figure 2F: the WB of DCC and UNC5B do show clearly a cleavage fragment of ~50KDa. However, there is a 2-fold increase of DCC levels and 3-fold increase of UNC5B levels in netrin-1 KO, suggesting that the cleavage increase is the result of higher levels of proteins, and not more cleavage as stated in the text. Some levels of DCC fragment are detectable in the netrin-1 WT blots. To show that DCC cleavage (and UNC5B) is increased in absence of netrin-1, the ratio of cleaved DCC/DCC should be quantified.

Minor comment: the identification of the figure 2 and supplemental fig. 1 are mislabeled in the result section.

3) Figure 3: the controls adenovirus +shDCC or shUNC5B are missing. It would be important to show that depleting DCC or UNC5B in the presence of netrin do not have major effects on dopamine cell survival.

4) 3E: the rescue effect of unc5b are quite modest as mentioned in the result section. However, the explanation on page 6 suggests that it might be because UNC5B is poorly expressed in SN. According to the WB (fig3d and 2f), the UNC5B levels are reasonable in the SN lysates. There is no data in the paper suggesting the UNC5B levels are poor in SN. One explanation could be that UNC5B is less efficient to rescue the cell death (3C and 3D), which correlates with being less efficient to rescue the motor deficits. What about shDCC+shUNC5B? will the effects be increased? The rescue of both DCC and UNC5B alone is modest, it is possible that both receptors are required to mediate cell death and the motor deficits in the absence of netrin-1.

5) The authors conclude that DCC cleavage may sensitize neurons to degeneration in the absence of netrin-1. In my opinion, this is not demonstrated in fig3. First, as mentioned above, it is not clear to me that the loss of netrin increases DCC or UNC5B cleavage but it is clear that the loss of netrin increases the protein levels. the authors show clearly that the loss of netrin increases cleaved caspase3, but they don't show that DCC or UNC5B are cleaved by caspases, as suggested in supp fig S2. If the authors used an anti-DCC antibody against the c-terminus of DCC (as mentioned in Mat and methods), the 50 KDa fragment suggest that the cleavage occurs at the extracellular side, probably by the metalloproteases as shown in previous publications. More work needs to be done to show that DCC cleavage sensitize neurons to degeneration. Protease inhibitors or caspase inhibitors could help to resolve these issues. One alternative possibility is that DCC and UNC5B signals together by heterodimerization for example to induce cell death.

Minor comment: fig3b is not mentioned in the text, page 6.

6) Supp fig4d and 4E: the authors mentioned that netrin-1 elicited neurite outgrowth by measuring the proportion of dopamine neurons in culture and increased TH surface area. This is not the proper way to do it, the authors should show the neurons in culture and measure the neurite length.

7) Supp 4F: why the DCC levels are reduced? This is not mentioned in the result section.

8) Fig6a: how many patient samples? What is DCC and UNC5B gene expression profile in this microarray dataset? 6b: quantification is from how many brain samples, n=3?

Minor comment: it is not clear to me why in some figures the stats are done with SD and in others with SEM.

Referee #3:

Jasmin et al. presents the neuroprotective and neurorestorative role of netrin-1 overexpression in mouse and rat models of PD. Although the role of netrin-1 and its receptor DCC is extensively studied in the development of mammalian CNS, their role in adult brain is not well known. Given the fact that netrin-1 is highly expressed in dopaminergic neurons of SN which are selectively vulnerable to degeneration in Parkinson's disease, this manuscript shows that loss of Netrin-1 and increased DCC cleavage cause motor impairments and loss of dopaminergic neurons in SN of adult mice. Moreover, the authors demonstrate that these phenotypes can be reversed by the overexpression of netrin-1 or brain administration of recombinant netrin-1. They ultimately extend their findings to the human pathology and show that netrin-1 expression is significantly decreased

in the SN of PD patients compared to control patients.

While the manuscript presents a potentially promising therapeutic approach and contains plenty of *in vivo* data, there are serious shortcomings at the moment that lower my overall enthusiasm. Most importantly, the mechanism behind their approach remains unsolved.

- 1- The authors show that netrin-1 is highly expressed in adult SN and is reduced in PD patients compared to control patients. They propose that the loss of netrin-1 might be a reason for the vulnerability and neurodegeneration of SN in PD patients. However, additional experiments are needed that further support this conclusion. Since dopaminergic neurons in ventral tegmental area show similar netrin-1 expression patterns as in SN but are less vulnerable in neurodegeneration. What is the effect of netrin-1 loss in such less vulnerable dopaminergic neurons?
- 2- Pacelli et al. (2015) showed that reducing axonal arborization by acting on axon guidance pathways reduces the vulnerability of SN Dopaminergic neurons to neurotoxic agents. Since, netrin-1 and DCC also have a role in neurite outgrowth, investigating if netrin-1-DCC dependent neurite outgrowth is involved in the vulnerability of SN dopaminergic neurons would help to start providing insight into the molecular mechanisms involved in the netrin-DCC pathway dependent vulnerability.
- 3- What is the rationale behind the experiment in fig 1d? Can this be explained more clearly?
- 4- In fig 2, besides the increase of cleaved DCC also full length DCC is increased in Netrin-1 KO mice compared to WT mice. Moreover, in fig 3, the reduction of DCC results in the reduction of cleaved DCC but also full length DCC and rescues the TH+ cell loss in netrin KO mice, however the author's conclusions are only based on the lower cleaved DCC levels. How is the effect of cleaved and full length DCC dissociated given they are both lowered in levels? The investigation of the role of full length DVV in dopaminergic vulnerability would also be appropriate/needed.
- 5- The PD animal models are sub-optimal: the toxins artificially induce DA death, but it is not clear if their mechanism of action and the mechanisms at play in PD are the same. Over expression of synuclein is also not optimal. It may be good to include a clear statement in the text as to the limitations of these models. Moreover, in Figure 4: what is the effect of MPTP on alpha-synuclein, netrin and DCC levels in wild type and SNCA mice?
- 6- Netrin-1 and GDNF have very similar effects on behavior, TH+ fiber density and TH+ cell count. It raises the question whether they affect the same pathways. Does netrin-1 or GDNF injection in netrin-1 KO mice also rescue the neurodegeneration? Similarities/differences of these two substances should be further investigated.

Minor points:

In Fig 1b both sections show netrin-1 expression in mice midbrain. Indicate which of the two sections are more anterior.

In Fig 1d, the zoom region of DAPI, netrin-1 and DAT stainings are not indicated on the first netrin-1 image.

Page 5 line 12: mentions Supplemental Fig. S2 a,c however supp. Fig. S2a doesn't represent the mentioned data and there is no S2c. The authors want to refer Fig 2 a, c. Please correct.

Cre injection was done into SN of 3-month-old mice. Nigral neuron population and mice motor behavior were monitored six weeks after netrin-1 KO. That means animals were at least 4.5 months old. However, the titles of fig2 a and b indicates 3 months old.

In Fig 2b the difference between Cre virus and CTL virus is not obvious. Indications of the TH+

fibers with arrows would help the readers.

Fig 2f says N=3. Does N=3 mean 3 different animals or the experiment repeated 3 independent time with pooled tissue from the same animals?

In Fig 2f, labels of bar plot indicates DCC and UNC5B, but they don't specify whether it is FL or cleaved.

Fig3d bar plots: statistics are missing.

Fig 4a indicates tamoxifen induction at second week however it is not mentioned in the other figures when tamoxifen was supplied.

In Fig 4d, figure is missing 'ns' sign.

Fig 5c indicate in the figure which is the 6-OHDA lesion side like in figure 5d.

Fig 5d doesn't show the control side of the brain and there is no quantification of the control in Fig 5d.

Dear Editor,

Thank you very much for your kind evaluation of our manuscript. We really appreciate the referees' efforts to help us improve the quality of our manuscript entitled "The pair netrin-1/DCC regulates dopamine neuronal cell survival, death and impacts on Parkinson's disease". We have endeavored to address the comments made by the two referees and we believe that the resulting revision is now more compelling.

Reviewer #1:

Overall, this is a very interesting study that uncovers novel functions for netrin-1 and its receptor DCC in adult dopamine neurons, suggesting a potential therapeutic role of these proteins in PD treatment. In general, the data are convincing, well performed with appropriate controls. The quantification of the data and statistics are properly done in the majority of the figures. However, I disagree with some important conclusions of the paper and will explain my reasons below. In addition, there are some issues that need to be clarified, as detailed in the comments below:

We thank the referee for his/her very kind and positive comment and as suggested we revised some conclusions and have bringing additional data to further make the manuscript more convincing.

« 1) Figure 1: it would be nice to show a detailed staining of DCC and UNC5B in the SN regions as done for netrin-1 in Figure 1B.»

DCC and UNC5B immunostainings on rat SN slices are now shown in Fig S1A of the revised manuscript.

« 2) Figure 2F: the WB of DCC and UNC5B do show clearly a cleavage fragment of ~50KDa. However, there is a 2-fold increase of DCC levels and 3-fold increase of UNC5B levels in netrin-1 KO, suggesting that the cleavage increase is the result of higher levels of proteins, and not more cleavage as stated in the text. Some levels of DCC fragment are detectable in the netrin-1 WT blots. To show that DCC cleavage (and UNC5B) is increased in absence of netrin-1, the ratio of cleaved DCC/DCC should be quantified »

We agree that this point was not addressed carefully. Indeed, upon netrin-1 deletion we observed increased levels of DCC and UNC5B and increased levels of a 50-kDa receptor cleavage fragment (Fig 2F). As pointed by referee 1, whether the cleavage fragment increase is the result of more cleavage or higher protein levels was not clearly addressed although we also showed that titrating netrin-1 clearly induces DCC cleavage and cell death in neurons, *in vitro* (Fig S2 A and B). As suggested, we now provide two different loading of the gels for DCC and UNC5B immunoblot in addition to a quantification of the gels in Figure 2F. As the referee can now more clearly see, when looking to intensity of the FL DCC band in netrin-1 Wt in the 120ug lysate compared to the intensity of the FL DCC band in netrin-1 KO in the 60ug lysate, this somehow the same intensity, while looking at the cleavage band, it is barely detectable in the 120ug lysate netrin-1 wt, while much more present in the 60ug lysate in netrin-1 KO. This nicely supports the view that upon netrin-1 withdrawal DCC (and UNC5B) are cleaved.

« Minor comment: the identification of the figure 2 and supplemental fig. 1 are mislabeled in the result section »

We thank the referee for noticing this error, the identification of the Fig 2 and supplemental Fig 1 is now labeled properly.

« 3) Figure 3: the controls adenovirus +shDCC or shUNC5B are missing. It would be important to show that depleting DCC or UNC5B in the presence of netrin do not have major effects on dopamine cell survival »

As suggested, we performed the CTL adenovirus vector + ShDCC or CTL adenovirus vector + ShUNC5B virus vector transductions on DIV 10 primary cultures. The results are included as Supplementary Fig 2E. As expected according the dependence receptor paradigm, in the presence of netrin-1, we found that deletion of either DCC or UNC5B or both did not trigger neuronal apoptosis.

« 4) 3E: the rescue effect of unc5b are quite modest as mentioned in the result section. However,

the explanation on page 6 suggests that it might be because UNC5B is poorly expressed in SN. According to the WB (fig3d and 2f), the UNC5B levels are reasonable in the SN lysates. There is no data in the paper suggesting the UNC5B levels are poor in SN. One explanation could be that UNC5B is less efficient to rescue the cell death (3C and 3D), which correlates with being less efficient to rescue the motor deficits. What about shDCC+shUNC5B? will the effects be increased? The rescue of both DCC and UNC5B alone is modest, it is possible that both receptors are required to mediate cell death and the motor deficits in the absence of netrin-1.»

We agree with the referee and has according modulated the text of our manuscript. Regarding expression of UNC5B vs DCC. UNC5B signals by immunoblot or immunostaining are globally weaker than DCC signals. Although those are not quantitative methods and that the differences in signal intensity may be due to a technical bias or difference of antibody affinity, what led us to speculate that UNC5B levels are poor in the SN, is that we had to load twice as much lysates to obtain reasonable signals for UNC5B by immunoblot (120µg of lysate) than for DCC immunoblot (60µg) while in most our other “cancer” settings UNC5B immunoblots show clean UNC5B expression. This is also supported by literature. Indeed even though there is very little literature on UNC5B in the SN compared to DCC, we checked UNC5B (Annex 1A) and DCC (Annex 1B) expression in the SN of adult mouse by ISH, using the Allen Brain Atlas. We can see a marked signal for DCC ISH (Annex 1B), clearly localized in the SN region (arrow) while UNC5B ISH signals (Annex 1A) in the SN are less obvious (Nissl coloration shows the tissue architecture and is used as an internal control (Annex 1C, D)). Altogether, and although gene expression does not always reflect protein expression, this body of observations led us to hypothesized that DCC may play a more preponderant role than UNC5B in the adult SN leading us to focus on DCC. However, we do not exclude that UNC5B alone may be indeed less efficient in inducing cell death than DCC or that both receptors may be required. This is now mentioned in the revised version of the manuscript (page 6).

Annex 1 UNC5B vs DCC expression by ISH in the adult mouse brain (from the Allen Brain Atlas)

« 5) The authors conclude that DCC cleavage may sensitize neurons to degeneration in the absence of netrin-1. In my opinion, this is not demonstrated in fig3. First, as mentioned above, it is not clear to me that the loss of netrin increases DCC or UNC5B cleavage but it is clear that the loss of netrin increases the protein levels. the authors show clearly that the loss of netrin increases cleaved caspase3, but they don't show that DCC or UNC5B are cleaved by caspases, as suggested in supp fig S2. If the authors used an anti-DCC antibody against the c-terminus of DCC (as mentioned in Mat and methods), the 50 KDa fragment suggest that the cleavage occurs at the extracellular side, probably by the metalloproteases as shown in previous publications. More work needs to be done to show that DCC cleavage sensitize neurons to degeneration. Protease inhibitors or caspase inhibitors could help to resolve these issues. One alternative possibility is that DCC and UNC5B signals together by heterodimerization for example to induce cell death.»

We thank the referee for this important point. First in response to previous comment 2, we now more adequately show that netrin-1 silencing is associated with more DCC (and UNC5B) cleavage. Regarding the fragment migrating at 50kDa, as mentioned in Mat and methods, this cleavage fragment was indeed detected using an anti-DCC antibody against the C-terminus domain of DCC which could suggest that it includes the all intracellular domain rather than the “caspase-cleaved fragment “1291-C terminus”. However we and other have repeatedly in the past observed that the caspase cleavage of both DCC and UNC5B were migrating to higher “size” (Tang et al., 2008)). To more formally determine whether this fragment migrating at 50kDa is due to caspase cleavage we treated DIV 10 dopamine neurons with caspase 3 and pan-caspase inhibitors and used a biologic (4Fbn) to titrate netrin-1. Blocking caspases’ activation by these inhibitors prevented netrin-1 deprivation-induced DCC or UNC5B proteolytic cleavage and loss of tyrosine hydroxylase (TH) levels (Supplemental Fig S2D).

In addition and to more formally implicate the DCC caspase cleavage in the degeneration effect described in the manuscript, we have now performed the behavior test in 6-OHDA model in mice point mutated for the caspase cleavage site of DCC. As the referee may be aware we have been developing a knock-in mice for which the endogenous DCC locus was mutated to generate a DCC mutated in D1290 (mutation D1290N). This mouse model was shown specifically shown to have DCC-induced apoptosis silenced and were shown to be more prone to develop cancer (Castets et al., Nature, 2012; Broutier et al., EMBO MM, 2016; Boussouar et al., Cancer Res, 2020). As shown in the Annex 2 below the specific inactivation of DCC-induced cell death was associated with a significant improvement in the behavior at 2weeks and with a clear trend toward an increase of TH+ fibers density. This supports further the implication of DCC caspase cleavage/pro-death activity, even though, at this stage we cannot discard that not only DCC-induced cell death is involved in the effect seen with netrin-1 silencing. This is not more adequately discussed in the manuscript

Annex. 2. Preventing DCC-induced cell death is neuroprotective in a mouse 6- OHDA model of PD. (A)

Experimental design. DCC^{mut/mut} and WT were lesioned with 6-OHDA in two sites of the right striatum (2 x 1µg). Amphetamine-induced behaviour was assessed at week two post lesion. At week 6 brains were taken to perform immunohistochemical (IHC)analyses. **(B)** Amphetamine-induced rotations at two weeks after lesion. Data are shown as individual values and mean ± SEM. n = 11 each group. Tukey post hoc analysis after two-way ANOVA, *P < 0.05 compared to vehicle group. **(C)** Density of TH-positive fibres in the striatum at six weeks post lesion. Individual values and mean ± SEM are shown, n=11 in each group. Unpaired t-test.

« Minor comment: fig3b is not mentioned in the text, page 6.»

Figure 3B is now mentioned in the text.

« 6) Supp fig4d and 4E: the authors mentioned that netrin-1 elicited neurite outgrowth by measuring the proportion of dopamine neurons in culture and increased TH surface area. This is not the proper way to do it, the authors should show the neurons in culture and measure the neurite length.»

This was indeed not correct. We are not including the analysis of netrin-1 induced outgrowth in dopamine neurons in vitro because this has been performed and published before with convincing results (Lin *et al*, 2005; Xu *et al*, 2010). However, as we show for the first time that in vivo netrin-1 (genetic overexpression or recombinant protein) (Fig 4C and 5C) is increasing the density of fibers, we have rephrased our conclusion to take support of the published in vitro works.

« 7) Supp 4F: why the DCC levels are reduced? This is not mentioned in the result section.»

We thank the referee for pointing this out. It is now mentioned in the text. Indeed, we show that briefly (4 hours) after netrin-1 injection, DCC levels are reduced while PTEN and PS6 levels are decreased and increased, respectively. The modulation of a receptor's expression level or cell surface presentation is a common mechanism for tuning responses to extracellular ligands. Netrin-1 exposure or loss can readily participate to modulate DCC levels and thus the cell response. In line with our results, several studies reported a fast and transient decrease in DCC levels in response to netrin-1 (DeGeer *et al*, 2013; Kim *et al*, 2005; Neuhaus-Follini & Bashaw,

2015) while netrin-1 KO mice exhibit increased levels of DCC (Bin *et al*, 2015).

The rapid decrease of DCC levels can occur through different processes. In *Xenopus*, the modulation of spinal neuron chemoattraction is mediated by the endocytosis and the slower protein synthesis of DCC (Ming *et al*, 2002). In rat embryonic cortical neurons, DCC is ubiquitinated, internalized and then degraded rapidly after netrin-1 exposure (Kim *et al*, 2005). Other reported that DCC proteolysis was required for guidance and that DCC cytoplasmic cleavage by gamma-secretase produced a DCC ICD fragment that may serve as a transcriptional coactivator in vertebrates (Bai & Pfaff, 2011; Neuhaus-Follini & Bashaw, 2015). Thus, it seems that the rapid regulation of DCC levels is a feature of neuronal and axonal remodeling in response to netrin-1.

« 8) Fig6a: how many patient samples? What is DCC and UNC5B gene expression profile in this microarray dataset?»

This dataset was obtained from 9 control patients and 16 PD patients. UNC5B gene expression is significantly increased in PD patients vs control patients. On the contrary DCC gene expression seems to be decreased in PD patients (Annex 3).

Annex 3: DCC and UNC5B expression levels in PD patients vs control patients from gds2821 dataset

« 6b: quantification is from how many brain samples, n=3?»

We used brain lysates from the substantia nigra of 3 healthy control patients and 3 PD patients (1 sample/well). Three technical replicates were performed by immunoblot. We have updated Figure 6 legend to make it clearer. Even though this number of individual is small, it is not “chosen” samples. The three independent PD brain samples were showing less netrin-1 than the three “normal” brain. Moreover, this is in line with the microarray dataset shown in Figure 6A.

« Minor comment: it is not clear to me why in some figures the stats are done with SD and in others with SEM.»

This is now more adequately explained. When value represents a different individual, individual values are shown, and data are expressed as mean \pm SD to show the variation among values. When means are compared, data are expressed as mean \pm SEM.

Reviewer #2:

“While the manuscript presents a potentially promising therapeutic approach and contains plenty of in vivo data, there are serious shortcomings at the moment that lower my overall enthusiasm. Most importantly, the mechanism behind their approach remains unsolved.”

We thank the referee for his/her positive comment. We have tried to answer experimentally the different important comments from the referee and hope the revised version of the manuscript is now more convincing.

« 1- The authors show that netrin-1 is highly expressed in adult SN and is reduced in PD patients compared to control patients. They propose that the loss of netrin-1 might be a reason for the vulnerability and neurodegeneration of SN in PD patients. However, additional experiments are needed that further support this conclusion. Since dopaminergic neurons in ventral tegmental area show similar netrin-1 expression patterns as in SN but are less vulnerable in neurodegeneration. What is the effect of netrin-1 loss in such less vulnerable dopaminergic neurons?»

We thank the referee for this very elegant suggestion. Substantia nigra (SN) and ventral tegmental area (VTA) dopamine (DA) neurons exhibit similar levels of netrin-1 but their sensitivity to netrin-1 seem to be different. Indeed, (Li *et al*, 2014) showed that midbrain dopamine axon attraction and elongation are induced at lower netrin-1 concentrations in SN pars compacta explants as compared to VTA explants. Also, nigral dopaminergic axons do not respond to high

netrin-1 levels contrary to VTA axons. These differences of sensitivity may be due to different receptor expression levels or patterns. For example, in the adult ventral midbrain DCC levels, which was shown to be expressed in both dopaminergic neurons of the VTA and SN, are the highest in the ventral tier dopamine neurons of the substantia nigra pars compacta (Osborne *et al*, 2005; Reyes *et al*, 2013) and therefore has been proposed to be a marker for this precise population of neurons.

To test whether SN and VTA neurons share similar sensitivity to netrin-1, we looked at the effect of the loss of netrin-1 in those regions. Indeed, in the previous version of the manuscript, we could see, in supplemental Fig S1 that after AAV-Cre injection in the SN (Netrin-1 KO), netrin-1 signals were significantly reduced in both the SN and in the VTA. In Figure 3, however, we could notice that upon netrin-1 KO, TH signals were lost in the SN and at a lesser extent in the VTA. We performed additional immunostaining on midbrain brain slices, comprising both the SN and VTA, that we have updated in the revised version (Fig 2 A and B). Clearly, dopamine neurons (TH-positive) from the SN, but not the VTA, are specifically lost upon netrin-1 deletion. VTA DA neurons are thus indeed more resistant to netrin-1 deprivation than SN DA neurons. This is, in our view, a very important observation that strengthens the specificity of our data and once again supports the overall notion of dependence receptor where the ability of these dependence receptors to trigger apoptosis in the ligand is really dependent on cellular context. This point is now discussed in the manuscript.

« 2- Pacelli et al. (2015) showed that reducing axonal arborization by acting on axon guidance pathways reduces the vulnerability of SN Dopaminergic neurons to neurotoxic agents. Since, netrin-1 and DCC also have a role in neurite outgrowth, investigating if netrin-1-DCC dependent neurite outgrowth is involved in the vulnerability of SN dopaminergic neurons would help to start providing insight into the molecular mechanisms involved in the netrin-DCC pathway dependent vulnerability.»

We completely agree with this important comment. Pacelli et al. (2015) paper is indeed very interesting. Although the massive arborization of SN dopamine neurons likely contributes to their vulnerability (because the energetic cost is high) it is also key to their proper function *in vivo*. Dopamine neuron arborization is highly plastic in the control of motricity and it is indeed highly likely that the fine regulation of the levels of guidance cues is critical for their maintenance and plasticity (Lesnick *et al*, 2008; Lin *et al*, 2009).

Concerning netrin-1 levels and the vulnerability of SN dopamine neurons, it is interesting to note that the dorsal striatum, characterized by poor levels of netrin-1, is associated with the degeneration of medium spiny neurons in Huntington's disease and SN dopamine projections in Parkinson's disease. On the contrary, the ventral striatum, characterized by higher levels of netrin-1, is associated with striatal (dopamine) hyperfunction (schizophrenia, drug addiction). Netrin-1 levels may thus reflect the potential for reinnervation of dopamine neurons (Shatzmiller *et al*, 2008). Also, it appears that SN dopamine neurons are more sensitive to netrin-1 levels than VTA dopamine neurons (Li *et al*, 2014) (Fig 2 A and B).

Finally, we would like to stress that our paper deals mostly with adult dopamine neurons and most of the scientists working with this system are not able to culture adult DA neurons.

Therefore, we assessed the ability of netrin -1 to induce dopamine neuron neurite outgrowth after 6-OHDA lesion in adult animals performing *in vivo* experiments. In this manuscript we demonstrated for the first time that Netrin-1 is inducing an increase of fibers density in adult dopamine neurons *in vivo*, suggesting an increase of outgrowth. These data are presented in Fig 5C and 4C and fit well with the ability of netrin-1 to induce neurite outgrowth in embryonic dopamine neurons has been performed and published before (Lin *et al*, 2005; Xu *et al*, 2010). We have discussed these data in the revised version of the manuscript and balance more adequately our conclusion between a survival role of netrin-1 that we demonstrate here and a role in DA outgrowth that may also be of importance.

« 3- What is the rationale behind the experiment in fig 1d? Can this be explained more clearly? »

In Fig 1D we looked at netrin-1 levels in the striatum because it is the structure that contains SN dopamine axon terminals. Indeed, SN dopamine neurons form a diffuse and massive arborization in the striatum which is key for the proper regulation of basal ganglia function in the control of motricity, but which also accounts for their vulnerability since the maintenance and function of this arborization may be highly energy demanding. Thus, when it comes to study SN DA neuron homeostasis and function, it is important to consider and characterize their arborization, that is, the striatal fibers. For example, it was proposed that neurodegeneration in PD starts at axon terminals and progresses retrogradely to the cell body (the dying-back theory) (Burke & O'Malley, 2013). Neurorestoration studies on survival factors, such as Jnk and GDNF, have shown that rescuing neuron cell body was not enough, to achieve functional neurorestoration, axon terminals also must be restored (Ries *et al*, 2008; Kirik *et al*, 2000).

DCC is expressed in SN dopamine cell body but also at axon terminals, so we found it relevant to check whether netrin-1 is also present at axon terminals, to get insight into how netrin-1/DCC may contribute to SN DA neuron homeostasis. Following referee's request, we expanded on it in the revised manuscript.

« 4- In fig 2, besides the increase of cleaved DCC also full length DCC is increased in Netrin-1 KO mice compared to WT mice. Moreover, in fig 3, the reduction of DCC results in the reduction of cleaved DCC but also full length DCC and rescues the TH+ cell loss in netrin KO mice, however the author's conclusions are only based on the lower cleaved DCC levels. How is the effect of cleaved and full length DCC dissociated given they are both lowered in levels? The investigation of the role of full length DCC in dopaminergic vulnerability would also be appropriate/needed. »

We thank the referee for noting this important point. As suggested by the referee 1 and as answered to the referee 1, we now more convincingly show that netrin-1 silencing is associated with an increase of DCC caspase cleavage. We also provide additional evidence for the role of DCC caspase cleavage in DA neuron homeostasis (Figure 1. DCC D1290N mutant). However the referee 2 is correct in the sense that the caspase cleavage and DCC pro-apoptotic activity probably do not account for the all effect and this is more adequately explain in the text of the manuscript. However, if the referee is considering a "positive" role of DCC in triggering some kind

of signaling upon netrin-1 binding that would positively impact on DA fate, it has to be argued that it has probably not a preponderant (at least in the models used here): First silencing DCC (with or not UNC5B) per se does not have any obvious impact on primary neurons as shown in FigS2 but rather prevents the effect of netrin-1 silencing. Second silencing of DCC *in vivo* is not phenocopying netrin-1 silencing but rather rescues netrin-1 silencing as shown in Figure 3.

« 5- The PD animal models are sub-optimal: the toxins artificially induce DA death, but it is not clear if their mechanism of action and the mechanisms at play in PD are the same. Over expression of synuclein is also not optimal. It may be good to include a clear statement in the text as to the limitations of these models. Moreover, in Figure 4: what is the effect of MPTP on alpha-synuclein, netrin and DCC levels in wild type and SNCA mice? »

We agree with the referee that both neurotoxin as well as alpha-synuclein animal models of Parkinson's disease are not optimal. Since no naturally occurring animal forms of PD are known, the development of animal models of the disease has proved indispensable to study the pathology but also therapy for PD. However, PD is a complex disease of unknown etiology which renders it difficult to fully mimic. In most of the studies researchers use only one animal model. Being aware of the limitations of the models, we decided to use 3 independent models in this study, 6-OHDA, MPTP and alpha-synuclein and we hope that this could strengthen our conclusions. We now have discussed the limitations of animal models of PD in the revised manuscript (pages 7-8).

« 6- Netrin-1 and GDNF have very similar effects on behavior, TH+ fiber density and TH+ cell count. It raises the question whether they affect the same pathways. Does netrin-1 or GDNF injection in netrin-1 KO mice also rescue the neurodegeneration? Similarities/differences of these two substances should be further investigated. »

We thank the referee for this interesting point. We have used GDNF in these experiments as a control, as a golden standard much used in the animal models of PD. Of interest, there are several fundamental differences between the action of GDNF and Netrin-1 on dopamine neurons that are now briefly discussed in the revised version of the manuscript. Firstly, it is firmly established that GDNF signals in dopamine neurons solely via RET receptor (and not via alternative receptors such as NCAM or syndecan-3) triggering survival promotion and neurite outgrowth through PI3-kinase-AKT and MAPK pathways. Netrin-1 induces positive signals at least via 2 receptors DCC and UNC5B through diverse signaling pathway, however the death observed upon netrin-1 withdrawal is not due to a loss of survival pathways but rather due to engagement of caspase-9-induced cell death. Secondly, GDNF is not expressed by dopamine neurons but is expressed by parvalbumin positive neurons in the striatum. These neurons secrete GDNF that is taken up by dopamine neuron axons projecting to the striatum and GDNF receptor complex is retrogradely transported to the substantia nigra pars compacta. Unlike netrin-1 which is required for the maintenance of adult DA neurons, endogenous GDNF is not required for the survival of adult dopamine neuron of the substantia nigra, *in vivo* (J.Kopra, 2015). Indeed, the numbers of TH+

cells and brain dopamine levels are unchanged in mice GDNF conditional KO mice (Nestin-Cre or AAV5-Cre injected GDNF^{fl/fl} mice). Contrary to GDNF, netrin-1, as well as its receptors, are expressed by dopamine neurons and obviously act by autocrine loop, at least at the cell body level. Netrin-1 is expressed by striatal interneurons and may thus also act in a paracrine way at axon terminals. Importantly, netrin-1 is not retrogradely transported to the substantia nigra pars compacta. This is more adequately described in the text of the manuscript.

« Minor: In Fig 1b both sections show netrin-1 expression in mice midbrain. Indicate which of the two sections are more anterior.»

We thank the referee for this comment. We have now indicated which section is more anterior.

«Minor: In Fig 1d, the zoom region of DAPI, netrin-1 and DAT stainings are not indicated on the first netrin-1 image.»

The zoom region is now indicated in the revised figures

« Minor: Page 5 line 12: mentions Supplemental Fig. S2 a,c however supp. Fig. S2a doesn't represent the mentioned data and there is no S2c. The authors want to refer Fig 2 a, c. Please correct.»

Thank you for pointing this mistake, it has now been corrected.

«Minor: Cre injection was done into SN of 3-month-old mice. Nigral neuron population and mice motor behavior were monitored six weeks after netrin-1 KO. That means animals were at least 4.5 months old. However, the titles of fig2 a and b indicates 3 months old..»

Thank you for noticing it, legends and titles were updated.

« Minor: In Fig 2b the difference between Cre virus and CTL virus is not obvious. Indications of the TH+ fibers with arrows would help the readers.»

We added arrows where we injected the virus vector in Figure 2.

“Minor: Fig 2f says N=3. Does N=3 mean 3 different animals or the experiment repeated 3 independent time with pooled tissue from the same animals?”

We apologize that it was not clear, Fig 2F legend was updated.

“Minor: In Fig 2f, labels of bar plot indicates DCC and UNC5B, but they don't specify whether it is FL or cleaved.”

Fig 2F bar plot legend is now updated.

“Fig3d bar plots: statistics are missing.”

The statistics are now shown in Fig 3D

“Fig 4a indicates tamoxifen induction at second week however it is not mentioned in the other figures when tamoxifen was supplied.”

The model used in Fig4A-D is a mice line that expresses the tamoxifen-inducible Cre-ERT2 fusion protein to induce netrin-1 conditional expression. We used this model to test whether netrin-1 could rescue DA neuron in mice intoxicated with 6-OHDA. This is the only model in the paper using this tamoxifen-inducible Cre-ERT2 fusion protein system, therefore there is no mention of tamoxifen injection in the other figures.

“In Fig 4d, figure is missing 'ns' sign.”

Thank you for noticing it, the 'ns' sign is now present.

“Fig 5c indicate in the figure which is the 6-OHDA lesion side like in figure 5d.”

The 6-OHDA lesion side is the right side, the control side is the left. It is now indicated in the Figure.

“Fig 5d doesn't show the control side of the brain and there is no quantification of the control in Fig 5d.”

The quantification of the control is not shown since results are expressed as percent of the intact (control) side. We quantified the TH+ cell number in the lesion side and reported it to the TH+ cell number of the control side. The coronal slices used for the analysis and shown are from similar antero-posterior positions. A representative image of the control side is provided.

Dear Patrick,

Thanks for submitting your revised manuscript to The EMBO Journal. Your study has now been seen by referees # 1 and 3.

As you can see from the comments below while the referees appreciate that the analysis has been strengthened they also have some remaining questions regarding the downstream mechanisms. Referee #1 would like further input into the role of DCC cleavage in this process and also has some questions about the results of the rescue experiment in Fig 3B. Regarding figure 6B do you have any more patient samples? Referee #3 has some comments regarding the patient data on DCC expression.

Everyone agrees that the strength of the paper lies in the in vivo mouse models and the potential therapeutic angle, but would be good to clarify these last points in a final revision. I can accept if certain aspects of the downstream mechanism is not fully clear, but then make sure to discuss this in a good way.

When you submit the revised version, will you also take care of the following points

- We need 3-5 keywords
- We don't have senior author distinction - just corresponding authors. Please double check.
- Please upload high resolution individual figure files.
- The appendix file is missing a ToC and figures should be labelled as Appendix Figure S1, Appendix Table S1 etc. Please also fix callout in text
- Some of the images need scale bars like Figure 1B, 5C&D
- In the data availability section please only add data sets generated in this study. As far as I can see no data is generated that needs to be deposited in a database. If this is correct please state: This study includes no data deposited in external repositories
- For the reference list - for articles with more than 10 authors the author list should be cut after 10 authors followed by et al.
- The funding info should also be added to the online system
- The lower panel in Figure 2A looks like it has a cut in in
- Please upload source data for all blots (one PDF file per figure)
- I have asked our publisher to do their pre-publication checks on the paper. They will send me the file within the next few days and I will pass it to you.
- We include a synopsis of the paper (see <http://emboj.embopress.org/>). Please provide me with a general summary statement and 3-5 bullet points that capture the key findings of the paper.

- We also need a summary figure for the synopsis. The size should be 550 wide by [200-400] high (pixels). You can also use something from the figures if that is easier.

That should be all. Let me know if you need any further input from my side

With best wishes

Karin

Karin Dumstrei, PhD
Senior Editor
The EMBO Journal

Further information is available in our Guide For Authors:

The revision must be submitted online within 90 days; please click on the link below to submit the revision online before 20th Jan 2021.

Referee #1:

In this revised version, the authors did not address carefully the comments of the reviewers. In term of DCC cleavage, they are not showing a ratio of cleaved DCC/DCC. In fact, it is not very clear how they are presenting the data with numbers below zero. DCC can be cleaved by caspase as shown by the authors and other groups, however, it is not well supported in the present study that this is required for the biological effects observed in the KO netrin-1 or OE netrin-1 models. In general, the effects of KO netrin-1 or OE netrin-1 are significant and clear but the downstream mechanisms are not clearly presented in the study. For instance, the rescue experiments in Fig. 3E are not very convincing, the effects are very mild. In figure 6B, showing a western blot of only 3 control samples and 3 patients seems to be rather small.

Referee #3:

The authors addressed all of our comments elaborately and conducted additional experiments to strengthen their conclusions. The revised version of the manuscript will be more accessible for neuroscience researchers. One final comment that I would like to see addressed relates to Annex 3. The authors show that PD patients have reduced DCC expression values in microarray gds2821 datasets. However western blot data in figure 6B show increased levels of DCC in PD patients. Moreover, Annex 3 figures show different results, one significant and one not significant, for DCC expression. Can the authors comment on the contradictory results obtained with microarray versus western bot and explain the difference between the DCC expression bar graphs in Annex3.

Overall, the paper demonstrates a novel role of Netrin-1 and its receptor DCC in the context of cell-specific vulnerability in PD and the data are convincing. Although the manuscript doesn't reveal mechanism behind their approach, I believe it opens new research avenues on this topic.

Response to editorial comments:

- Following keywords has been added in the title page: Netrin-1/ DCC, neurorestoration, Parkinson's disease
- Senior author distinction has been added
- ToC has been added in the Appendix and figures has been labelled as Appendix Figure S1, Appendix Table S1 etc. The main text has been corrected
- Scale bars has been added in Figure 1B, 5C&D
- The statement "*This study includes no data deposited in external repositories*" has been added in the data availability section.
- The reference list format has been corrected
- The funding info is added to the online system
- The original data and specifically Figure 2A are provided. Source data for all blots (one PDF file per figure) has been uploaded
- Please find below a general summary statement and 3-5 bullet points that capture the key findings of the paper. We also added a summary figure.

Synopsis

This study shows, using various in vivo approaches, a novel role for netrin-1 as a key regulator of adult nigral dopamine neuron maintenance and highlights the therapeutic potential of targeting netrin-1 signalling in Parkinson's disease.

Bullet points:

- Netrin-1 is required for the maintenance of adult dopamine neurons in the substantia nigra.
- Loss of netrin-1 in the substantia nigra induces receptor cleavage and apoptotic cell death of dopamine neurons
- Netrin-1 is protective and neurorestorative in various models of Parkinson's disease

Reviewer #1:

In this revised version, the authors did not address carefully the comments of the reviewers. In term of DCC cleavage, they are not showing a ratio of cleaved DCC/DCC. In fact, it is not very clear how they are presenting the data with numbers below zero.

We apologize for not having fully satisfied referee 1 concerns in the former version. Indeed, whether the cleavage fragment increase is the result of more cleavage or higher protein levels was not clearly addressed. To answer this, we quantified the ratio of cleaved DCC/ full length (FL) DCC and cleaved UNC5B/FL UNC5B, as suggested by referee 1, and updated **Fig 2F** (quantification bar graph) accordingly. The quantification shows that the ratio of cleaved receptor/ FL receptor is increased in netrin-1 KO samples, demonstrating that the increase of the receptor cleavage fragments upon netrin-1 deletion is not only the results of higher levels of proteins but is also the result of more cleavage.

DCC can be cleaved by caspase as shown by the authors and other groups, however, it is not well supported in the present study that this is required for the biological effects observed in the KO netrin-1 or OE netrin-1 models.

We agree that we are showing here mainly correlations between DCC cleavage and the biological effects. We are not *per se* demonstrating that this cleavage is causally implicated and we have more carefully discussed this in the discussion of the manuscript “Additionally, simultaneous deletion of DCC partly prevented dopamine neuron loss, suggesting that DCC, via

its death-inducing activity, regulates DA neurons survival. However it is fair to say that we cannot exclude that not only the death activity of DCC is important to mediate this DA loss but also alternative loss of netrin-1-induced DCC signaling”.

In general, the effects of KO netrin-1 or OE netrin-1 are significant and clear but the downstream mechanisms are not clearly presented in the study. For instance, the rescue experiments in Fig. 3E are not very convincing, the effects are very mild.

The rescue experiments in Fig.3E are indeed mild but the data are significant. What we can definitely conclude is that silencing of DCC or UNC5B is NOT mimicking netrin-1 silencing, so, at least it demonstrates clearly that if there are downstream mechanisms, they are not ligand-induced receptor mediated downstream signaling. This being said, as noted previously we have included a statement in the discussion suggested that it may be both the pro-death activity and additional downstream signaling.

In figure 6B, showing a western blot of only 3 control samples and 3 patients seems to be rather small.

We agree with the referee. Unfortunately we did not have access to additional patient samples despite effort from the 3 groups involved. Again to downplay the data obtained we have modified the text accordingly: “these results in human, even though at this step preliminary because of limited access to materials, may indicate that the reduction of netrin-1 in patients could contribute to the progression of PD”.

Reviewer #3:

The authors addressed all of our comments elaborately and conducted additional experiments to strengthen their conclusions. The revised version of the manuscript will be more accessible for neuroscience researchers.

We thank referee 3 for her/his constructive comments that helped us to strengthen our conclusions.

One final comment that I would like to see addressed relates to Annex 3. The authors show that PD patients have reduced DCC expression values in microarray gds2821 datasets. However western blot data in figure 6B show increased levels of DCC in PD patients. Moreover, Annex 3 figures show different results, one significant and one not significant, for DCC expression. Can the authors comment on the contradictory results obtained with microarray versus western blot and explain the difference between the DCC expression bar graphs in Annex3.

We thank the referee for noting this point. Indeed in the WB using the three PD

patients/control patients, DCC was rather seen more detected in PD brains than in controls while the dataset analysis was either showing no change at the messenger level or rather a reduction of DCC. This goes with the comment from the referee 1 that led us to introduce a sentence in the text to limit the conclusion based on the human materials as we had only access to 3 PD specimens “these results in human, even though at this step preliminary because of limited access to materials, may indicate that the reduction of netrin-1 in patients could contribute to the progression of PD”. This being said, DCC gene is a huge gene located spanning in chromosome 18q and our experience in microarrays when looking at DCC, is that microarray data rarely predicts DCC mRNA expression and even less DCC protein (this is why we did not include the analysis of receptors in the manuscript). We additionally added the sentence in the main text: “Of note, the increase level of DCC/UNC5B was not seen using the above gene-expression profiling dataset.”.

We are grateful to the reviewers for their comments, which we believe have strengthened the manuscript.

If we should send additional information, please let me know. We thank you in advance for your consideration of the revised manuscript.

Dear Patrick,

Thanks for sending me the revised manuscript. I have now had a chance to take a look at it and I appreciate the introduced changes.

I am therefore very pleased to accept the manuscript for publication here.

with best wishes

Karin

Karin Dumstrei, PhD
Senior Editor
The EMBO Journal

Please note that it is EMBO Journal policy for the transcript of the editorial process (containing referee reports and your response letter) to be published as an online supplement to each paper. If you do NOT want this, you will need to inform the Editorial Office via email immediately. More information is available here: https://emboj.embopress.org/about#Transparent_Process

Your manuscript will be processed for publication in the journal by EMBO Press. Manuscripts in the PDF and electronic editions of The EMBO Journal will be copy edited, and you will be provided with page proofs prior to publication. Please note that supplementary information is not included in the proofs.

Should you be planning a Press Release on your article, please get in contact with embojournal@wiley.com as early as possible, in order to coordinate publication and release dates.

If you have any questions, please do not hesitate to call or email the Editorial Office. Thank you for your contribution to The EMBO Journal.

Corresponding Author Name: MEHLEN Patrick

Journal Submitted to: EMBO J

Manuscript Number: 2020-105537R